# Sensitive red protein calcium indicators for imaging neural activity

**Hod Dana[1]\*, Boaz Mohar[1,2], Yi Sun[1], Sujatha Narayan[1], Andrew Gordus[3], Jeremy P Hasseman[1], Getahun Tsegaye[1], Graham T Holt[1], Amy Hu[1], Deepika Walpita[1], Ronak Patel[1], John J Macklin[1], Cornelia I Bargmann[3], Misha B Ahrens[1], Eric R Schreiter[1], Vivek Jayaraman[1], Loren L Looger[1], Karel Svoboda[1], Douglas S Kim[1]\***

[1]Janelia Research Campus, Howard Hughes Medical Institute, Ashburn, United States; [2]Weizmann Institute of Science, Rehovot, Israel; [3]Howard Hughes Medical Institute, The Rockefeller University, New York, United States

**Abstract** Genetically encoded calcium indicators (GECIs) allow measurement of activity in large populations of neurons and in small neuronal compartments, over times of milliseconds to months. Although GFP-based GECIs are widely used for in vivo neurophysiology, GECIs with red-shifted excitation and emission spectra have advantages for in vivo imaging because of reduced scattering and absorption in tissue, and a consequent reduction in phototoxicity. However, current red GECIs are inferior to the state-of-the-art GFP-based GCaMP6 indicators for detecting and quantifying neural activity. Here we present improved red GECIs based on mRuby (jRCaMP1a, b) and mApple (jRGECO1a), with sensitivity comparable to GCaMP6. We characterized the performance of the new red GECIs in cultured neurons and in mouse, *Drosophila*, zebrafish and *C. elegans* in vivo. Red GECIs facilitate deep-tissue imaging, dual-color imaging together with GFP-based reporters, and the use of optogenetics in combination with calcium imaging.

**\*For correspondence:** danah@ janelia.hhmi.org (HD); kimd@ janelia.hhmi.org (DSK)

## Introduction

Genetically-encoded calcium indicators (GECIs) enable non-invasive measurement of neuronal activity in vivo. Activity can be tracked across multiple spatial scales, from synapses (*Chen et al., 2013b*) to populations of thousands of neurons (*Ahrens et al., 2013*; *Peron et al., 2015*). Neuronal dynamics can be probed over times of milliseconds (*Chen et al., 2013a*; *O'Connor et al., 2013*; *Li et al., 2015*) to months (*Huber et al., 2012*; *Margolis et al., 2012*). Green fluorescent protein (GFP)-based GECIs (*Tian et al., 2009*; *Akerboom et al., 2012*; *Ohkura et al., 2012*; *Chen et al., 2013b*), such as GCaMP6 (*Chen et al., 2013b*), are widely used for imaging neural activity. GCaMP6 indicators exhibit excellent signal-to-noise ratio, allowing detection of single action potentials (APs) in many situations (*Chen et al., 2013b*; *Peron et al., 2015*).

Applications of GCaMPs and other widely used GECIs are limited by their excitation and emission spectra. GCaMPs are difficult to use in transgenic animals that already express other GFP-based proteins. The blue excitation light used in standard wide-field microscopy can cause photodamage and is highly scattered in tissue. The green GCaMP emission is absorbed by blood (*Svoboda and Block, 1994*), which reduces the penetration depth of imaging in vertebrates in vivo. In addition, the GCaMP excitation spectrum overlaps with those of light-sensitive ion channels, including channelrhodopsin-2 (ChR2) (*Nagel et al., 2003*), which complicates the simultaneous use of green GECIs and optogenetics. Red-shifted GECIs thus promise three main advantages over GFP-based sensors: increased maximal imaging depth, parallel use of a red GECI with light-sensitive ion channels for all-

**eLife digest** Neurons encode information with brief electrical pulses called spikes. Monitoring spikes in large populations of neurons is a powerful method for studying how networks of neurons process information and produce behavior. This activity can be detected using fluorescent protein indicators, or "probes", which light up when neurons are active.

The best existing probes produce green fluorescence. However, red fluorescent probes would allow us to see deeper into the brain, and could also be used with green probes to image the activity and interactions of different neuron types simultaneously. However, existing red fluorescent probes are not as good at detecting neural activity as green probes.

By optimizing two existing red fluorescent proteins, Dana et al. have now produced two new red fluorescent probes, each with different advantages. The new protein indicators detect neural activity with high sensitivity and allow researchers to image previously unseen brain activity. Tests showed that the probes work in cultured neurons and allow imaging of the activity of neurons in mice, flies, fish and worms.

History has shown that enhancing the techniques used to study biological processes can lead to fundamentally new insights. In the future, Dana et al. would therefore like to make even more sensitive protein indicators that will allow larger networks of neurons deeper in the brain to be imaged.

optical neurophysiology experiments, and reduced photodamage. In addition, together with existing green GECIs, red GECIs allow simultaneous imaging of multiple components of neuronal circuitry.

Current red GECIs share overall architecture with the GCaMP sensors. They are based on circularly permuted red fluorescent proteins (RFPs), a calcium-binding protein (calmodulin) and a binding peptide (M13 or ckkap). RCaMP1 (*Akerboom et al., 2013*) is derived from mRuby (*Kredel et al., 2009*), whereas R-GECO (*Zhao et al., 2011*; *Wu et al., 2014*) and R-CaMP2 (*Inoue et al., 2015*) are derived from mApple (*Shaner et al., 2008*). R-GECO is more sensitive than RCaMP1. However, mApple-based GECIs, such as R-GECO and R-CaMP2, exhibit photoswitching when illuminated with blue light, causing a transient increase of red fluorescence that complicates their use in optogenetics (*Akerboom et al., 2013*; *Wu et al., 2013*). Here we performed large-scale structure-guided mutagenesis and neuron-based screening (*Wardill et al., 2013*) to develop improved red GECIs, starting with RCaMP1h (*Akerboom et al., 2013*) and R-GECO1 (*Zhao et al., 2011*). We report on the mRuby-based jRCaMP1a and jRCaMP1b, and mApple-based jRGECO1a, all of which show several-fold improved sensitivity for detecting neural activity compared to their parent scaffolds.

## Results

### Protein engineering

R-GECO1 and RCaMP1 are based on circularly permuted mApple (*Shaner et al., 2008*) and mRuby (*Kredel et al., 2009*), respectively, fused to calmodulin (CaM) and the CaM-interacting M13 peptide (*Crivici and Ikura, 1995*). In the presence of calcium, CaM undergoes a conformational change and associates with M13 to form a complex proximal to the chromophore inside the RFP β-barrel (*Akerboom et al., 2009*; *2013*). The conformational change modifies the chromophore environment, modulating solvent access, chromophore $pK_a$, absorption, and quantum yield, and altogether results in increased RFP brightness. Structure-guided mutagenesis and screening in a neuron-based assay have been successful in improving GCaMP sensitivity and kinetics (*Akerboom et al., 2012*; *Chen et al., 2013b*). Here we applied a similar approach to red GECIs.

We focused mutagenesis on the interfaces between the RFP and CaM, between CaM and M13, and within CaM itself (78/442 and 87/451 amino acid positions were mutated to near saturation in RCaMP1h and R-GECO1, respectively, see *Supplementary files 1–2*) (*Figure 1a*). These regions are structurally homologous to the regions previously explored in the engineering of GCaMP6 (*Chen et al., 2013b*). Single-mutation variants (934 RCaMP1h; 689 R-GECO1, *Supplementary files 1–2*) were tested in an automated neuronal assay (*Wardill et al., 2013*) (*Figure 1b–d*). Dissociated

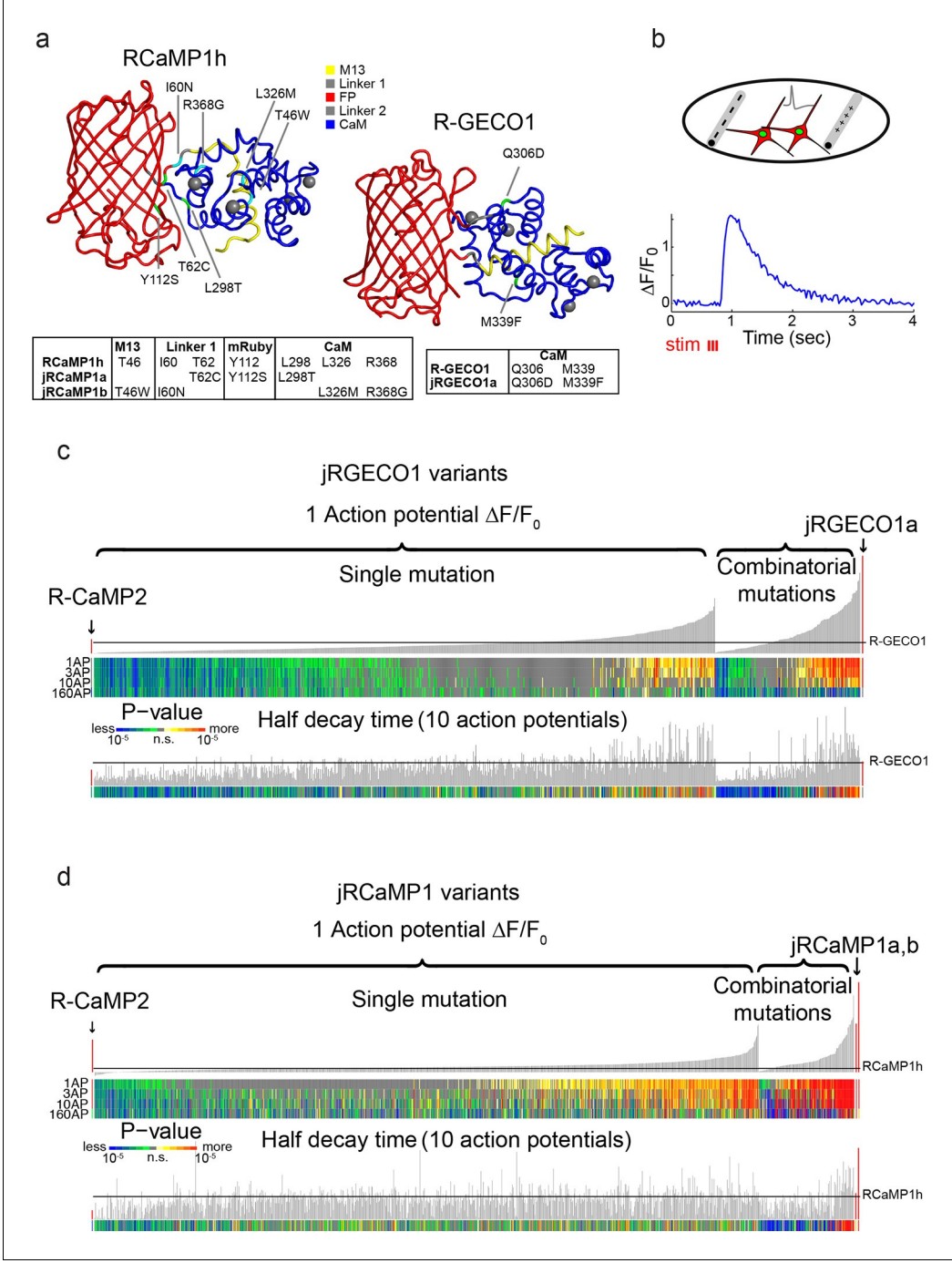

**Figure 1.** Mutagenesis and screening of jRCaMP1 and jRGECO1 in dissociated neurons. (**a**) RCaMP1h and R-GECO1 structure and mutations introduced in jRCaMP1a, jRCaMP1b, and jRGECO1a. M13 peptide (yellow), linker 1 (gray), cpmRuby or cpmApple (red), linker 2 (gray), CaM (blue). Mutation positions for jRCaMP1a (green), jRCaMP1b (cyan), jRGECO1a (green). (**b**) Schematic of the cultured neuron assay. Field electrodes (gray, upper panel) stimulate cultured neurons expressing a cytosolic red GECI variant and nuclear GFP. Changes in fluorescence are recorded (lower panel) and analyzed. An example response trace of a jRGECO1a-expressing neuron after 3 action potential (AP) stimulus is shown. (**c**) Screening results for 855 R-GECO1 variants. Top, fluorescence changes in response to 1 action potential (vertical bars, $\Delta F/F_0$ amplitudes; black bars, single R-GECO1 mutations and combinatorial mutations; red bars, R-CaMP2 left, jRGECO1a right). Middle, significance value for different AP stimuli (color plot). Bottom, half decay times after 10 APs. Black line indicates R-GECO1 performance levels. (**d**) Screening results for 1070 RCaMP1h variants. Top, fluorescence changes in response to 1 AP (same order as in **b**; red bars, R-CaMP2 left, jRCaMP1a and jRCaMP1b right). Middle, significance value for

*Figure 1 continued on next page*

*Figure 1 continued*

different AP stimuli (color plot). Bottom, half decay times after 10 APs. Black line indicates RCaMP1h performance levels.

rat hippocampal neurons in 96-well plates were transfected with plasmids expressing red GECI variants (RCaMP1h variant or R-GECO1 variant). To normalize for expression level, the plasmid also expressed GFP that was localized to the nucleus. R-GECO1 and RCaMP1h were both expressed throughout the cytoplasm and nucleus (data not shown). To sense calcium selectively in the cytoplasm we added a nuclear export sequence (NES) to the N-termini of R-GECO1, RCaMP1h, and all their variants. As expected, addition of the NES restricted expression to the cytoplasm (*Figure 1b*). R-CaMP2 was excluded from the nucleus without the NES.

A field electrode triggered trains of action potentials (APs) in all neurons within each well (Materials and methods, [*Wardill et al., 2013*]). Time-lapse images (800 μm x 800 μm fields of view; 35 Hz) were acquired before, during and after stimulation. Fluorescence changes were extracted from single neurons to compute the sensitivity, dynamic range, and kinetics of the responses to various trains of APs. Individual red GECI variants were compared to the parent constructs and published GECIs, including R-CaMP2 (*Inoue et al., 2015*), GCaMP6s, and GCaMP6f (*Chen et al., 2013b*).

Numerous single mutations (353/934 RCaMP1h; 187/689 R-GECO1, *Supplementary files 1–2*) improved sensitivity compared to the parent proteins (higher $\Delta F/F_0$ amplitude in response to one AP; $p < 0.01$, Wilcoxon rank sum test) (*Figure 1c,d*). For example, T46W, I60N, and I60T exhibited enhanced sensitivity and accelerated kinetics compared to RCaMP1h. I109K had similar effects on R-GECO1. M339F accelerated kinetics but did not affect response amplitude of R-GECO1. Beneficial mutations were sorted based on improved response amplitudes (in response to trains of 1, 3, and 10 APs) and/or faster kinetics, without a significant reduction in the maximal fluorescence change elicited by 160 APs.

Beneficial mutations were combined in a second round of mutagenesis (136 RCaMP1h and 166 R-GECO1 variants) (*Figure 1c,d*). Based on criteria similar to those outlined above, two new mRuby-based sensors, jRCaMP1a and jRCaMP1b, and one mApple-based sensor, jRGECO1a, were selected for in-depth analysis (red bars in the histograms of *Figure 1c,d*). These sensors have similar absorption and emission spectra to each other and their parent constructs but they differ in sensitivity for detecting neural activity, kinetics, and other biophysical properties (−*Figure 2*, *Figure 2—figure supplement 1–2*, *Figure 2—source data 1*).

jRGECO1a is the most sensitive indicator, with 8.5-fold larger $\Delta F/F_0$ amplitude for 1AP stimuli and faster rise time than R-GECO1 (*Figure 2*). Decay time and maximal $\Delta F/F_0$ amplitude were similar to R-GECO1. The improved amplitude and faster rise kinetics were associated with a tighter apparent calcium affinity (*Figure 2—source data 1*). jRGECO1a response amplitudes and kinetics are comparable to GCaMP6f for brief trains of 1–10 APs (*Chen et al., 2013b*). Similar to other mApple-based indicators, jRGECO1a exhibits photoswitching in response to blue light (*Figure 2—figure supplement 3*).

jRCaMP1a and jRCaMP1b were also much improved compared to their parent sensors (24-fold and 13-fold improved sensitivity for detecting 1 AP stimuli) (*Figure 2*). jRCaMP1a has higher sensitivity and slower decay kinetics than jRCaMP1b. jRCaMP1b has a larger dynamic range, without saturation in the range of 1–160 APs (*Figure 2c*). Both jRCaMP1a and jRCaMP1b have higher calcium affinities than RCaMP1h (*Figure 2—source data 1*). However, jRCaMP1a has very limited dynamic range in cultured neurons and solution (160 ± 25% and 320 ± 10% respectively, mean ± s.d.). jRCaMP1a and jRCaMP1b are two-fold brighter than jRGECO1a in the calcium-bound state (*Figure 2—figure supplement 2*). Similar to RCaMP1h, jRCaMP1a and jRCaMP1b did not exhibit photoswitching (*Figure 2—figure supplement 3*). This allows independent photostimulation and imaging in neurons that co-express ChR2 and jRCaMP1a/b (*Figure 2—figure supplement 4*).

## Red protein calcium indicators in mouse V1

We tested jRGECO1a, jRCaMP1a, jRCaMP1b, their parent indicators, and R-CaMP2 (*Inoue et al., 2015*) in the mouse primary visual cortex (V1) in vivo (*Chen et al., 2013b*) (*Figure 3a, Video 1*). The

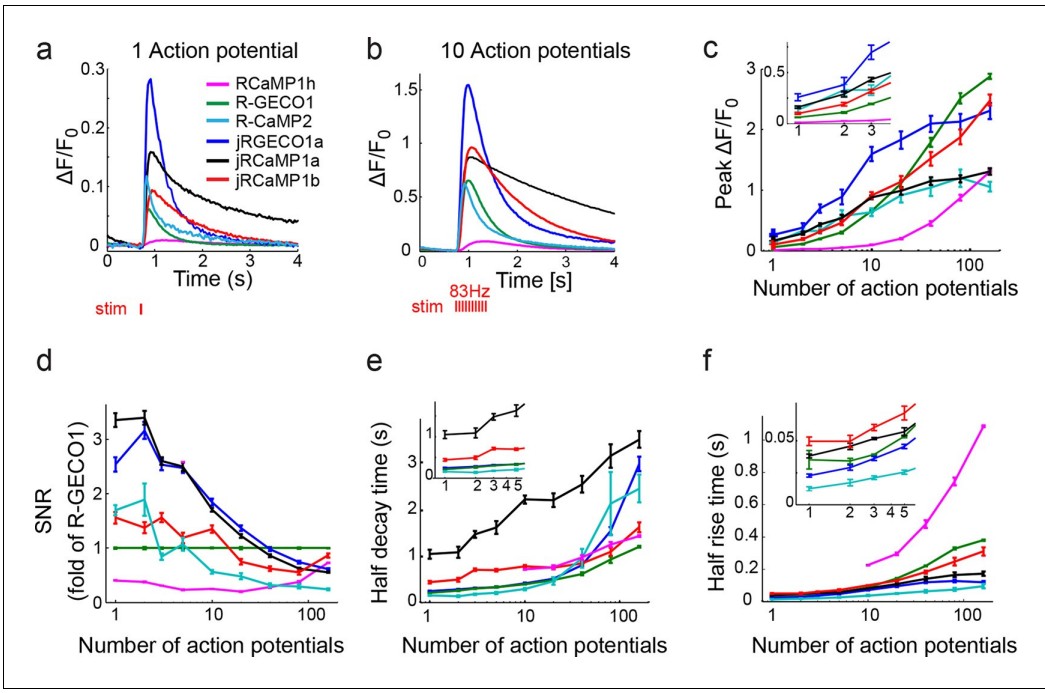

**Figure 2.** jRGECO1 and jRCaMP1 performance in dissociated neurons. (**a**) Average responses in response to one action potential (AP) for RCaMP1h (9479 neurons, 605 wells), R-GECO1 (8988 neurons, 539 wells), R-CaMP2 (265 neurons, 22 wells), jRGECO1a (383 neurons, 26 wells), jRCaMP1a (599 neurons, 38 wells), and jRCaMP1b (641 neurons, 31 wells). (**b**) Same for 10 APs response. (**c–f**) Comparison of jRGECO1 and jRCaMP1 sensors and other red GECIs, as a function of number of APs (color code as in **a**). (**c**) Response amplitude, $\Delta F/F_0$. (**d**) Signal-to-noise ratio, SNR, defined as the fluorescence signal peak above baseline, divided by the signal standard deviation before the stimulation is given. (**e**) Half decay time. (**f**) Half rise time. Error bars correspond to s.e.m (n=605 wells for RCaMP1h; 539, R-GECO1; 22, R-CaMP2; 38, jRCaMP1a; 31, jRCaMP1b; 26, jRGECO1a).

The following source data and figure supplements are available for figure 2:

**Source data 1.** Biophysical properties of purified jRGECO1 and jRCaMP1 sensors.
**Figure supplement 1.** Absorption and emission spectra of red GECIs.
**Figure supplement 2.** Biophysical properties.
**Figure supplement 3.** Photoswitching in purified protein assay.
**Figure supplement 4.** jRCaMP1a is more compatible than jRGECO1a for simultaneous use with ChR2.

majority of V1 neurons can be driven to fire action potentials in response to drifting gratings (**Mrsic-Flogel et al., 2007**; **Niell and Stryker, 2008**). V1 neurons were infected with adeno-associated virus (AAV) expressing one of the red GECI variants under the human synapsin1 promoter (AAV-SYN1-red GECI variant) and imaged 16–180 days later. Two-photon excitation was performed with a tunable ultrafast laser (Insight DS+; Spectra-Physics) running at 1040 nm or 1100 nm. L2/3 neurons showed red fluorescence in the neuronal cytoplasm. Visual stimuli consisted of moving gratings presented in eight directions to the contralateral eye (**Akerboom et al., 2012**; **Chen et al., 2013b**). Regions of interest corresponding to single neurons revealed visual stimulus-evoked fluorescence transients that were stable across trials and tuned to stimulus orientation (**Figure 3b**). Orientation tuning was similar for all constructs tested (**Figure 3—figure supplement 1**). Fluorescence transients tracked the dynamics of the sensory stimuli (**Figure 3b–d**, **Video 1**). mApple-based indicators tracked more faithfully than mRuby-based indicators because of their faster kinetics (signal half-decay time after end of stimulus was 300 ± 22 ms for R-GECO1, 175 cells; 390 ± 20 ms, jRGECO1a, 395 cells; 330 ±

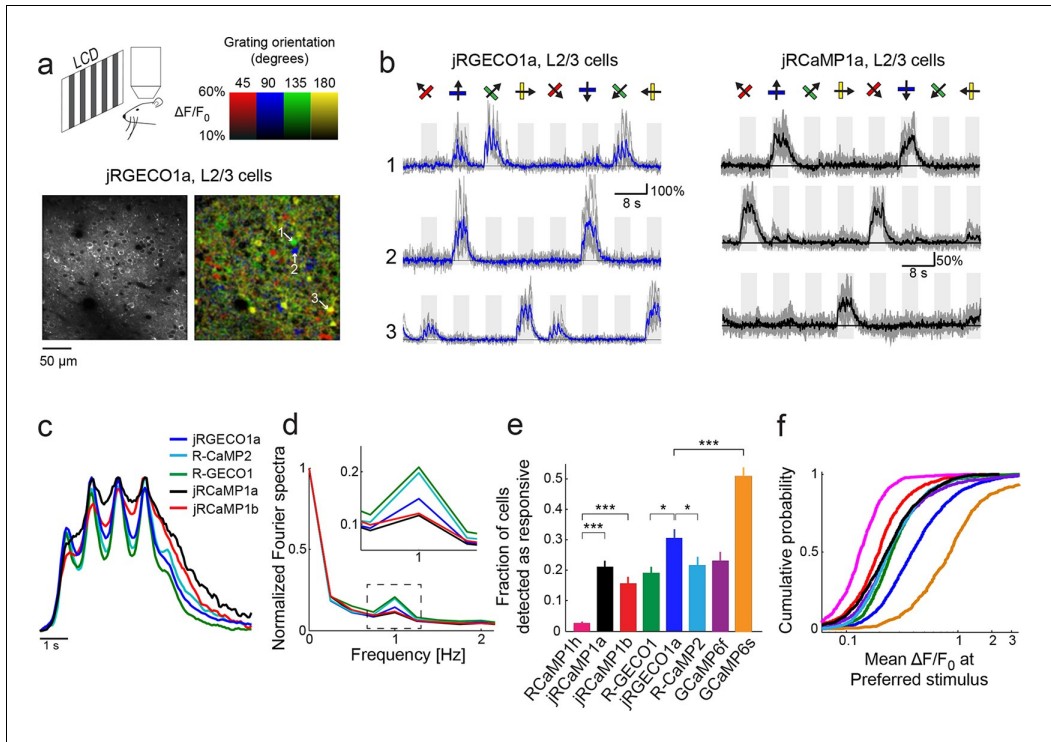

**Figure 3.** jRGECO1a and jRCaMP1a and jRCaMP1b performance in the mouse primary visual cortex. (**a**) Top, schematic of the experiment. Bottom, image of V1 L2/3 cells expressing jRGECO1a (left), and the same field of view color-coded according to the neurons' preferred orientation (hue) and response amplitude (brightness). (**b**) Example traces from three L2/3 neurons expressing jRGECO1a (left) and jRCaMP1a (right). Single trials (gray) and averages of 5 trials (blue and black for jRGECO1a and jRCaMP1a respectively) are overlaid. Eight grating motion directions are indicated by arrows and shown above traces. The preferred stimulus is the direction evoking the largest response. jRGECO1a traces correspond to the cells indicated in panel **a** (see also *Video 1*). (**c**) Average response of neurons to their preferred stimulus (175 cells, R-GECO1; 310, R-CaMP2; 395, jRGECO1a; 347, jRCaMP1a; 95, jRCaMP1b. n=4 mice for jRGECO1a and jRCaMP1a, n=3 mice for all other constructs. Panels c-f are based on the same data set. (**d**) Fourier spectra normalized to the amplitude at 0 Hz for neurons driven with 1 Hz drifting gratings, transduced with RCaMP1h, R-GECO1, R-CaMP2, jRGECaMP1a, jRCaMP1b, and jRGECO1a. Inset, zoomed-in view of 1 Hz response amplitudes. (**e**) Fraction of cells detected as responding to visual stimulus (ANOVA test, p<0.01) when expressing different calcium indicators. This fraction was 8- and 6-fold higher for jRCaMP1a and jRCaMP1b compared to RCaMP1h, respectively, and 60% higher for jRGECO1a compared to R-GECO1 (Wilcoxon rank sum test; *, p<0.05; **, p<0.01; ***, p<0.001). Error bars correspond to s.e.m (26 fields-of-view, RCaMP1h; 45, jRCaMP1a; 31, jRCaMP1b; 30, R-GECO1; 40, jRGECO1a; 33, R-CaMP2; 23, GCaMP6s; 29, GCaMP6f) (**f**) Distribution of ΔF/F amplitude for the preferred stimulus. A right-shifted curve, such as jRGECO1a vs. R-GECO1 or jRCaMP1a/b vs. jRCaMP1h, indicates enhancement of response amplitude (75 percentile values of 0.36 and 0.27 vs. 0.18 for jRCaMP1a and jRCaMP1b vs. RCaMP1h, and 0.66 vs. 0.38 for jRGECO1a vs. GCaMP6f, respectively). (1210 cells, R-GECO1; 861, RCaMP1h; 1733, R-CaMP2; 1605, jRGECO1a; 1981, jRCaMP1a; 971, jRCaMP1b; 907, GCaMP6f; 672, GCaMP6s), same colors as in **e**.

The following figure supplements are available for figure 3:

**Figure supplement 1.** Comparison of orientation tuning in V1 neurons measured with different red GECIs.

**Figure supplement 2.** Long-term expression of red GECIs in mouse V1.

16 ms, R-CaMP2, 310 cells; 640 ± 30 ms, jRCaMP1a, 347 cells; 500 ± 45 ms, jRCaMP1b, 95 cells; activity of RCaMP1h expressing cells was to weak to be reliably characterized, mean ± s.e.m., Materials and methods).

We compared the performance of red GECIs using standard metrics (*Chen et al., 2013b*). One measure of sensitivity is the fraction of neurons detected as responsive in the visual cortex

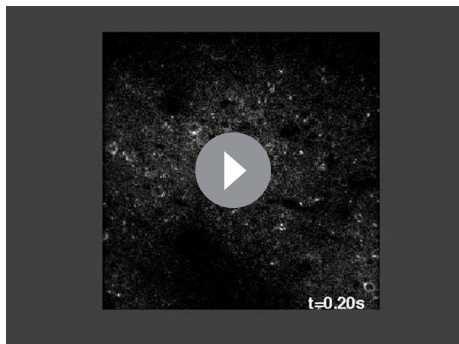

t=0.20s

**Video 1.** jRGECO1a L2/3 functional imaging in the mouse V1. The mouse was anesthetized and presented with moving gratings in eight directions to the contralateral eye. Gratings were presented for 4 s (indicated by appearance of an arrowhead in the grating propagation direction) followed by a 4 s of blank display. Field of view size was 250x250 μm², acquired at 15 Hz and filtered with a 5 frame moving average.

(*Figure 3e*). For jRCaMP1a and jRCaMP1b this fraction was 8- and 6-fold higher than for RCaMP1h ($p<10^{-6}$, Wilcoxon rank-sum test), and comparable to GCaMP6f. For jRGECO1a, the fraction was 60% higher than for R-GECO1 (p=0.012), 40% higher than R-CaMP2 (p=0.03), 30% higher than GCaMP6f (p=0.1), but 40% lower than GCaMP6s ($p<10^{-4}$). The mean $\Delta F/F_0$ at the preferred visual stimulus (Materials and methods) also showed enhanced sensitivity for jRCaMP1a and jRCaMP1b compared to RCaMP1h (*Figure 3f*). jRGECO1a responses in vivo were larger than other red GECIs and GCaMP6f, but still smaller than GCaMP6s responses.

Long-term and high expression of GCaMP was reported to affect neuronal health (*Tian et al., 2009*; *Chen et al., 2013b*). GCaMP6 nuclear-filled cells show attenuated responses to sensory stimuli, slower decay times, and degraded orientation selectivity (*Tian et al., 2009*; *Chen et al., 2013b*). We expressed jRGECO1a, jRCaMP1a, and jRCaMP1b for more than 130 days in mouse V1. Unlike GCaMP indicators, nuclear filling was not observed (*Figure 3—figure supplement 2a*), presumably due to the NES in the red GECI sequences. Red GECIs showed stable performance over time, although a small decrease of orientation selectivity was observed for jRCaMP1b after 180 days of expression (*Figure 3—figure supplement 2b–d*). Interestingly, long-term expression with the tested red GECIs seems to be more stable than with GCaMP6s (*Chen et al., 2013b*). These data suggest that jRGECO1a and jRCaMP1a / jRCaMP1b can be used in experiments requiring long-term expression.

The measured jRGECO1a and jRCaMP1a responses in the visual cortex were smaller than expected based on measurements in cultured neurons (*Figure 2a*, *Figure 4c*) and lower than those produced by GCaMP6s (*Chen et al., 2013b*). High-resolution microscopy of fixed brain tissue sections revealed bright fluorescent punctae in neurons labeled with jRGECO1a but not with jRCaMP1a/b. jRGECO1a punctae co-localized with LAMP-1, a marker of lysosomes (*Figure 4—figure supplement 1a*) (*Katayama et al., 2008*). Similar punctae were also found in cultured neurons, but were less numerous (Wilcoxon rank sum test, p<0.001, *Figure 4—figure supplement 1b*). When imaged in vivo, ROIs containing punctae had higher baseline fluorescence by 90% but lower peak $\Delta F/F_0$ by 20% compared to their surrounding soma (10 cells and 39 punctae, *Figure 4—figure supplement 1c*). This implies that accumulation of fluorescent, non-responsive jRGECO1a in lysosomes reduces the signal-to-noise ratio for in vivo imaging.

We also detected green fluorescence (500–550 nm) in the in vivo images of all red GECIs imaged in V1. Green fluorescence was visible with 900 nm excitation, but not with 1040 nm excitation, suggesting that red GECIs partition into at least two species. The green fluorescence was distributed unevenly across the cytoplasm and does not report calcium (*Figure 4—figure supplement 2*, *3a–b*). Indeed, jRGECO1a neurons with larger fractional fluorescence intensity at 900 nm showed lower response amplitudes. Longer expression times for jRGECO1a increased relative intensity at 900 nm and similarly correlated with lower peak $\Delta F/F_0$ (*Figure 4—figure supplement 3c–d*, F test, p-value<0.002). These data suggest that long-term expression in the mammalian brain can degrade sensitivity of the red GECIs.

Imaging with red probes suffers less from scattering of excitation light and absorption of fluorescence compared with GFP-based sensors (*Figure 2—figure supplement 1*), which could allow deeper imaging in vivo (*Horton et al., 2013*). To estimate how fluorescence decays with imaging depth we imaged apical dendrites from layer (L) 5 cells (*Figure 5a*). We fitted an exponential decay to the fluorescence signal measured as a function of depth. Red GECI fluorescence decayed much more slowly (130 ± 30 μm; median ± s.d., 19 dendrites, n=3 mice infected with RCaMP1h, R-

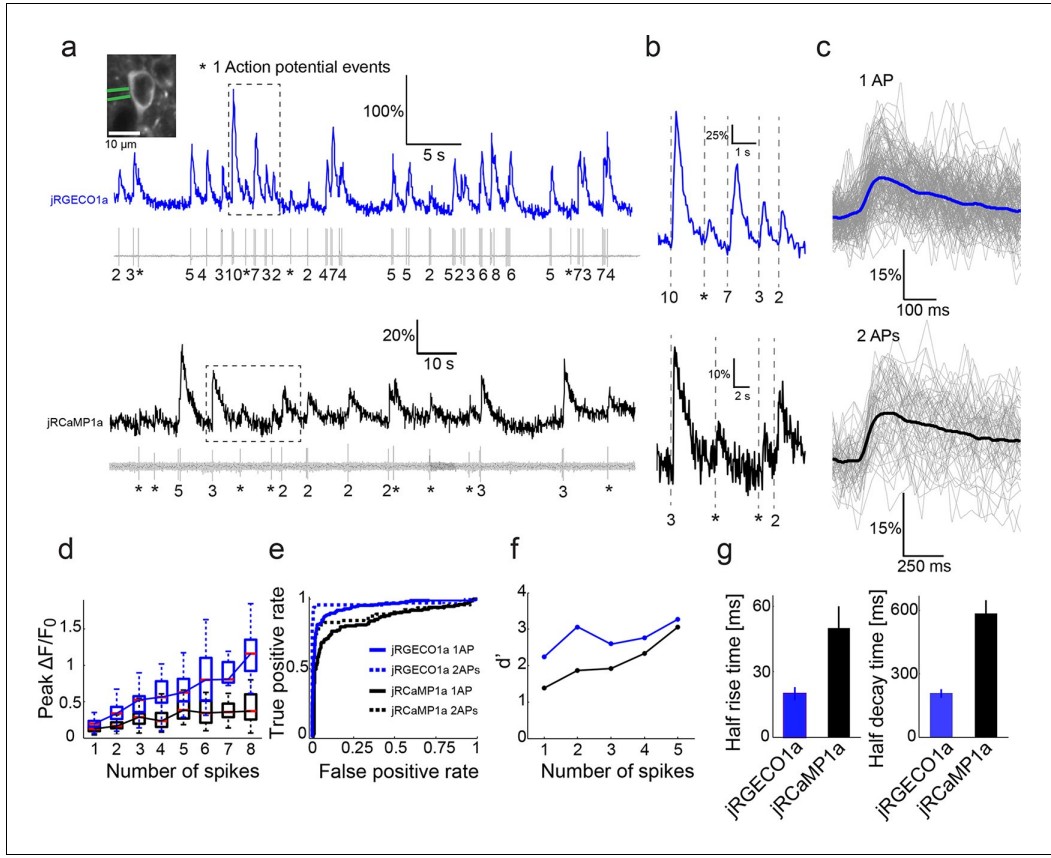

**Figure 4.** Combined imaging and electrophysiology in the mouse visual cortex. (**a**) Simultaneous fluorescence dynamics and spikes measured from jRGECO1a (top, blue) and jRCaMP1a (bottom, black) expressing neurons. The number of spikes for each burst is indicated below the trace (single spikes are indicated by asterisks). Left inset, a jRCaMP1a expressing neuron with the recording pipette (green). (**b**) Zoomed-in view of bursts of action potentials (corresponding to boxes in **a**). Top, jRGECO1a; bottom, jRCaMP1a. (**c**) jRGECO1a fluorescence changes in response to 1 AP (top, 199 spikes from 11 cells, n=6 mice), and jRCaMP1a fluorescence changes in response to 2 APs (bottom, 65 spikes from 10 cells, n=5 mice). Blue (top) and black (bottom) lines are the median traces. (**d**) Distribution of peak fluorescence change as a function of number of action potentials in a time bin (jRGECO1a, blue boxes: 199 1AP events; 2 APs: 70 events within a 100 ms time bin; 3 APs: 29, 125 ms; 4 APs: 34, 150 ms; 5 APs: 35, 175 ms; 6 APs: 22, 200 ms; 7 APs:14, 225 ms; 8 APs: 21, 250 ms. jRCaMP1a, black boxes: 135 1 AP events; 2 APs: 65, 150 ms; 3 APs: 71, 200 ms; 4 APs: 52, 250 ms; 5 APs: 33, 300 ms; 6 APs: 20, 350 ms; 7 APs: 14, 350 ms; 8 APs: 11, 350 ms). Each box corresponds to the 25th to 75th percentile of the distribution ($q_1$ and $q_3$ respectively), whisker length is up to the extreme data point or $1.5_l$ ($q3 - q1$). (**e**) Receiver operating characteristic (ROC) curve for classifying 1 and 2 APs for jRGECO1a and jRCaMP1a (jRGECO1a: 320 events of no AP firing within a 4 s bin, jRCaMP1a: 274 events of no AP firing within a 5 s bin, 1 AP and 2 APs data same as in **d**). (**f**) Detection sensitivity index (d') as a function of number of spikes in a time bin (same parameters as in **d–e**). (**g**) Comparison of mean half rise (left) and decay (right) times of jRGECO1a and jRCaMP1a for 2 AP response. Error bars correspond to s.e.m.

The following figure supplements are available for figure 4:

**Figure supplement 1.** jRGECO1a accumulates in lysosomes.

**Figure supplement 2.** Red GECIs lack functional response at 900nm excitation.

**Figure supplement 3.** Complex spectral and functional characteristics of the red GECIs after long-term expression in vivo.

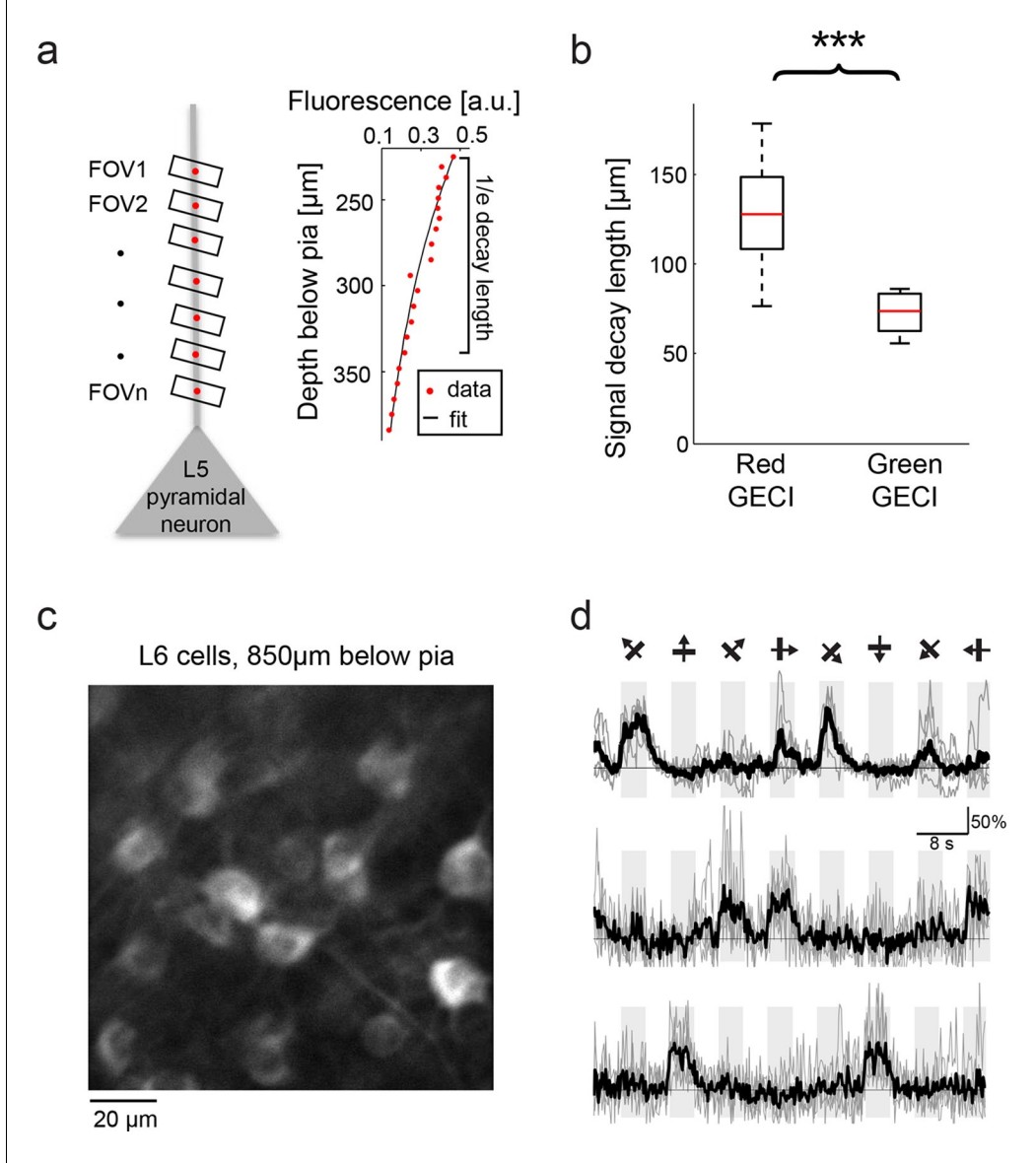

**Figure 5.** Deep tissue imaging using red GECIs. (a) Left, schematic of the measurement. L5 neuron apical dendrites were imaged at different depths (FOVs 1-n). Right, RCaMP1h fluorescence from an L5 apical dendrite (red dots) as a function of imaging depth. For fixed excitation light the brightness decreases as a function of imaging depth because of scattering and absorption losses. The decay was characterized by fitting an exponential function to the signal (solid black line). (b) Exponential decay coefficients measured from dendrites expressing green (GCaMP6s or GCaMP6f) or red (RCaMP1h, jRCaMP1a, and jRGECO1a) GECIs. Red GECI signal decay coefficients were significantly longer than for green GECI (Wilcoxon rank sum test, p<0.0001; 3 mice and 19 dendrites for red GECIs; 2 mice and 14 dendrites for green GECIs). (c) L6 neurons, 850 µm under the pia. An NTSR1-cre mouse (*Gong et al., 2007*) was infected with FLEX-SYN1-NES-jRCaMP1a AAV. (d) Example traces from three example L6 neurons. Single trials (gray) and averages of 5 trials (black) are overlaid. Eight grating motion directions are indicated by arrows and shown above traces.

GECO1, or jRGECO1a) than GCaMP6 fluorescence (75 ± 15 µm; median ± s.d., 14 dendrites, n=2 mice infected with GCaMP6s or GCaMP6f; p<0.0001, Wilcoxon rank sum test) (*Figure 5b*).

This enhanced imaging depth enabled, for example, imaging deep in L6. We infected L6 neurons by injecting AAV-SYN1-FLEX-jRCaMP1a virus into the visual cortex of *NTSR1*-cre (a L6-specific marker) mice (*Gong et al., 2007*). Four weeks after infection we could detect orientation-tuned

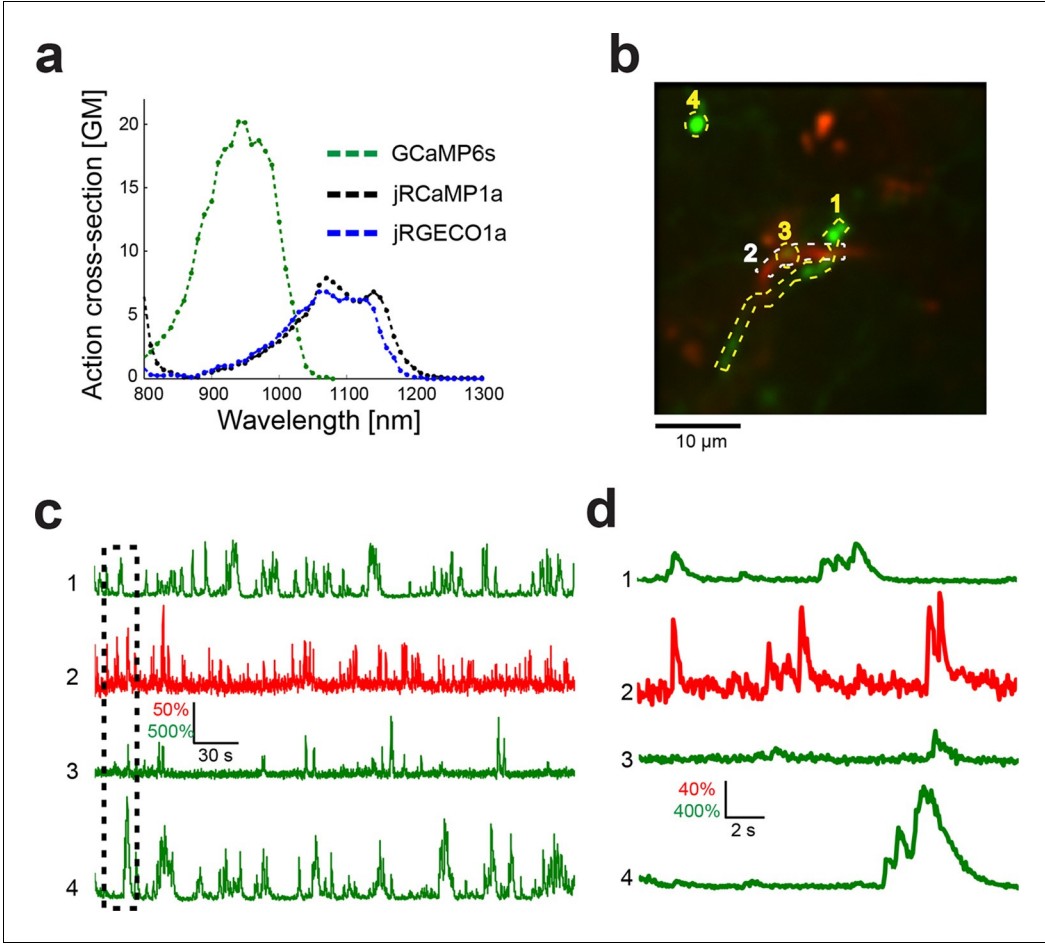

**Figure 6.** Dual color imaging in the mouse visual cortex. (a) Two-photon action spectra of Ca- saturated GCaMP6s, jRCaMP1a, and jRGECO1a. Measurements were done on purified protein (Materials and methods). (b) Image of L5 apical dendrites (red) and LM axons (green) imaged in L1 (50 μm under the pia) of KJ18-cre mice. (c) $\Delta F/F_0$ traces of axonal (green) and dendritic (red) ROIs, as indicated in b (*Video 2*). (d) Zoom-in corresponding to the dashed box in c.

somatic transients in L6 cells (down to 900 μm below the pia) in response to visual stimuli (*Figure 5c,d*).

The spectral properties of red GECIs allow dual-color imaging together with green GCaMP6 indicators. To demonstrate dual-color functional imaging in spatially intermingled neuronal processes, a FLEX-SYN1-jRGECO1a AAV was injected into V1 of mouse lines expressing cre recombinase specifically in L5 neurons (KJ18-cre [*Gerfen et al., 2013*]) and SYN1-GCaMP6s AAV was injected into cortical LM, which sends axons to V1 (*Andermann et al., 2011*; *Oh et al., 2014*). A single excitation source at 1000 nm was used to excite both indicators. Imaging was performed in L1 of V1, where signals from overlapping axons and dendrites could be reliably detected (*Figure 6*, *Video 2*). We note that bleaching of the red GECI was low when imaging relatively large cellular compartments, such as cell bodies or apical dendrites (*Video 1–2*), but significant bleaching was seen for thin axons. Larger compartments are more forgiving because of the dynamic balance between bleaching and diffusion of the imaged molecules.

## Relationship between spikes and fluorescence dynamics

We characterized the relationship between somatic fluorescence changes and spiking in L2/3 cells by combining imaging with loose-seal, cell-attached recordings (*Figure 4a,b*). The visual stimulus was adjusted online so that a large range of instantaneous firing rates was recorded for each cell.

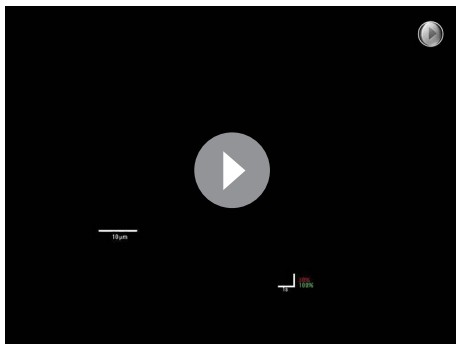

**Video 2.** Dual-color imaging of axons and apical dendrites in L1 of the mouse V1. jRGECO1a labeled apical L5 dendrites (red), and GCaMP6s labeled axons from LM (green) were imaged in L1 of V1 (left). $\Delta F/F_0$ traces from 4 ROIs (*Figure 6b*) are presented (right). Field of view size was 40x40 $\mu m^2$ acquired at 15 Hz and filtered with a 5 frame moving average.

Imaging was performed at high zoom to detect transients with the highest signal-to-noise ratio (*Peron et al., 2015*). jRGECO1a fluorescence changes allowed relatively robust detection of activity (*Figure 4c,d*). Single spikes were detected with 82% accuracy (5% false-positive rate, n=11 cells from 6 animals, 199 spikes) (*Figure 4e,f*). Two APs in a time bin of 100 ms were detected with 96% accuracy (5% false-positive rate, 70 events). Fluorescence changes of jRCaMP1a were smaller, with a detection rate for 1AP and 2AP events of 65% and 80% respectively (5% false-positive rate, n=10 cells from 5 animals, 135 and 65 events respectively). jRGECO1a showed higher $\Delta F/F_0$ amplitudes and better detection than jRCaMP1a over the entire range of AP bursts (*Figure 4d–f*). jRGECO1a also had faster kinetics (*Figure 4g*). Measured rise and decay time of jRGECO1a and jRCaMP1a were similar to GCaMP6f and GCaMP6s, respectively (*Chen et al., 2013b*).

## Red protein calcium indicators in *Drosophila*, zebrafish, and *C. elegans*

We next tested red GECIs in flies, zebrafish and worms. Red GECIs were expressed pan-neuronally in transgenic flies (R57C10-Gal4). Boutons were imaged in the larval neuromuscular junction (NMJ) after electrical stimulation (*Figure 7a,b*; Materials and methods) (*Hendel et al., 2008*). Consistent with cultured neuron data, we saw a significant boost of single AP sensitivity in jRGECO1a, jRCaMP1a, and jRCaMP1b variants compared to their parent indicators (p<0.008 for all comparisons; Wilcoxon rank sum test; *Figure 7c*, *Figure 7—figure supplement 1–2*, *Figure 7—source data 1–2*). Peak response amplitudes of jRGECO1a and jRCaMP1a outperform GCaMP6s in the 1–5 Hz stimulation range (single AP $\Delta F/F_0$ amplitude 11.6 ± 0.9%, 8.6 ± 0.5%, and 4.5 ± 0.3% respectively, mean ± s.e.m; *Figure 7c*, *Figure 7—figure supplement 3*, *Figure 7—source data 3*; 12 FOVs for jRGECO1a; 11, jRCaMP1a; 12, GCaMP6f; n=7 flies for all constructs). The decay kinetics of jRGECO1a (half decay time at 160 Hz: 0.42 ± 0.02 s, mean ± s.e.m) was similar to GCaMP6f (0.43 ± 0.01 s; *Figure 7d*, *Figure 7—figure supplement 3*, *Figure 7—source data 3*). The combination of high sensitivity and fast kinetics for jRGECO1a allows detection of individual spikes on top of the response envelope for stimuli up to 10 Hz (*Figure 7b*, *Figure 7—figure supplement 1*).

Red GECIs were also expressed pan-neuronally in transgenic zebrafish under the elavl3/HuC promoter (elavl3:GECI). Fish showing strong expression in the trigeminal (Tg) neurons 3–4 days post fertilization (dpf) were selected for imaging. Tg neurons are usually silent and fire one or few spikes in response to touch (*Douglass et al., 2008*). Brief trains of electrical stimulation pulses (20 ms each; 1, 5, and 10 pulses at 20 Hz), which are known to stimulate Tg neurons (*Akerboom et al., 2013*), were used to elicit responses in these cells and image calcium transients (*Figure 8a*; widefield one-photon imaging, Materials and methods). Similar to results from cultured rat hippocampal neurons, mice, and *Drosophila*, jRGECO1a was the most sensitive red GECI tested in this assay: its response to one and five pulses outperformed all other GECIs, including GCaMP6f and GCaMP6s (*Figure 8b,c*). For higher numbers of stimulus pulses the jRGECO1a response was partially saturated. jRCaMP1a and jRCaMP1b exhibited large improvements in sensitivity over their parent indicator, RCaMP1h, and also higher sensitivity than R-CaMP2, with jRCaMP1b showing faster kinetics than jRCaMP1a (*Figure 8d*).

Finally, Red GECIs were tested in the ASH and AWC neurons of *Caenorhabditis elegans*, two sensory neurons with distinct graded signaling properties (*Akerboom et al., 2012*; *Kato et al., 2014*). ASH neurons, which respond to noxious stimuli with calcium increases, were exposed to high-osmolarity glycerol in one second pulses alternating with buffer (*Figure 9a,b*). All tested GECIs exhibited an initial increase in fluorescence that was strongest for jRCaMP1b, jRCEGO1a, and R-GECO1 ($\Delta F/F_0$ amplitude ~250%, *Figure 9b*), followed by modulation of the fluorescence signal and a slow

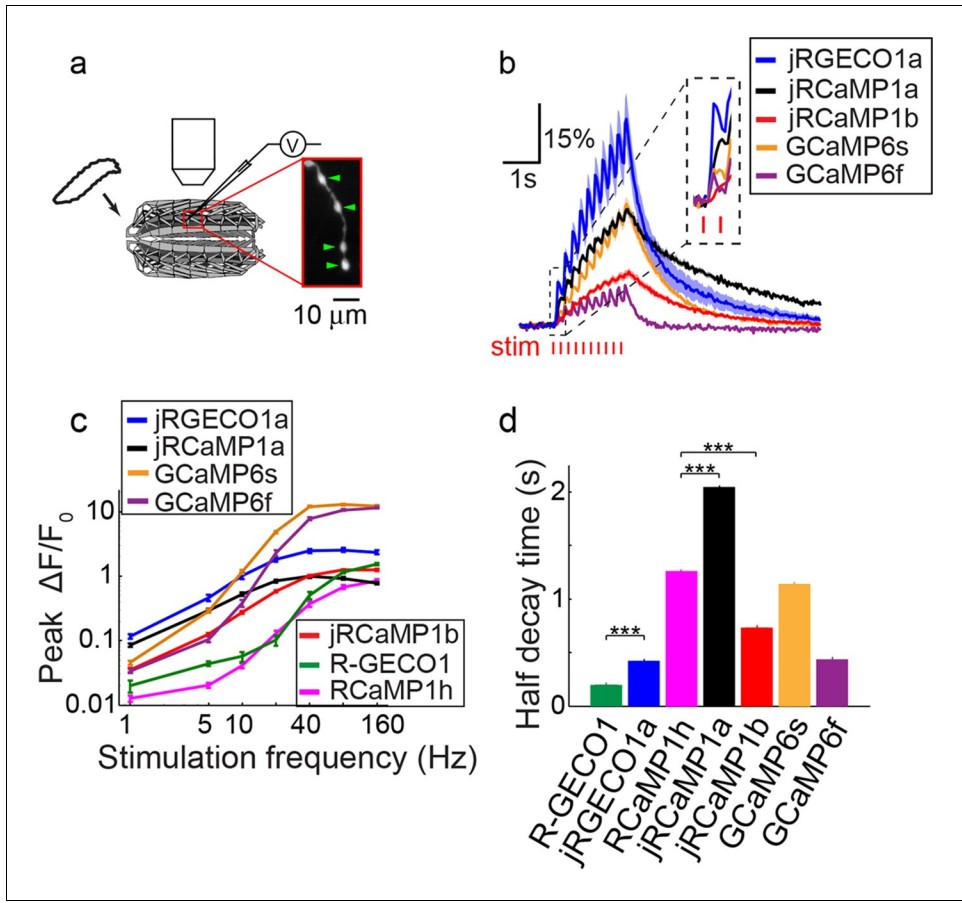

**Figure 7.** Imaging activity in *Drosophila* larval NMJ boutouns with red GECIs. (**a**) Schematic representation of *Drosophila* larval neuromuscular junction (NMJ) assay. Segmented motor nerve is electrically stimulated while optically imaging calcium responses in presynaptic boutons (green arrows). (**b**) Response transients (mean ± s.e.m.) to 5 Hz stimulation (2 s duration) for several red and green GECIs. Response amplitudes were 4-fold and 60% higher for jRGECO1a than GCaMP6f and GCaMP6s respectively ($p<10^{-4}$ and $p=0.01$, Wilcoxon rank sum test), jRCaMP1a response amplitude was 3-fold higher than GCaMP6f ($p=10^{-4}$) and similar to GCaMP6s (12 FOVs for jRGECO1a; 11, jRCaMP1a; 13, jRCaMP1b; 12, GCaMP6s; 12, GCaMP6f; n=7 flies for all constructs) (**c**) Comparison of frequency tuned responses (peak $\Delta F/F_0$, mean ± s.e.m.) of red and green GECIs for 1, 5, 10, 20, 40, 80 and 160 Hz stimulation (2 s duration). jRGECO1a and jRCaMP1a response amplitudes were 2–3 fold higher than GCaMP6s and GCaMP6f for 1 Hz stimulus ($p<0.001$, Wilcoxon rank sum test), but lower for stimulus frequencies of 20 Hz and higher (12 FOVs for jRGECO1a; 10, R-GECO1; 11, jRCaMP1a; 13, jRCaMP1b; 10, RCaMP1h; 12, GCaMP6s; 12, GCaMP6f; n=5 flies for R-GECO1, n=7 flies for all other constructs) (**d**) Half decay time (mean ± s.e.m.) of red and green GECIs at 160 Hz stimulation (same FOVs and flies as in **c**; ***, $p<0.001$, Wilcoxon rank sum test).

The following source data and figure supplements are available for figure 7:

**Source data 1.** Summary of results shown in *Figure 7—figure supplement 1*.

**Source data 2.** Summary of results shown in *Figure 7—figure supplement 2*.

**Source data 3.** Summary of results shown in *Figure 7—figure supplement 3*.

**Figure supplement 1.** Imaging activity in *Drosophila* larval NMJ boutons with jRGECO1a.

**Figure supplement 2.** Imaging activity in *Drosophila* larval NMJ boutons with jRCaMP1 constructs.

**Figure supplement 3.** Comparing red and green GECI activity in *Drosophila* larval NMJ boutons.

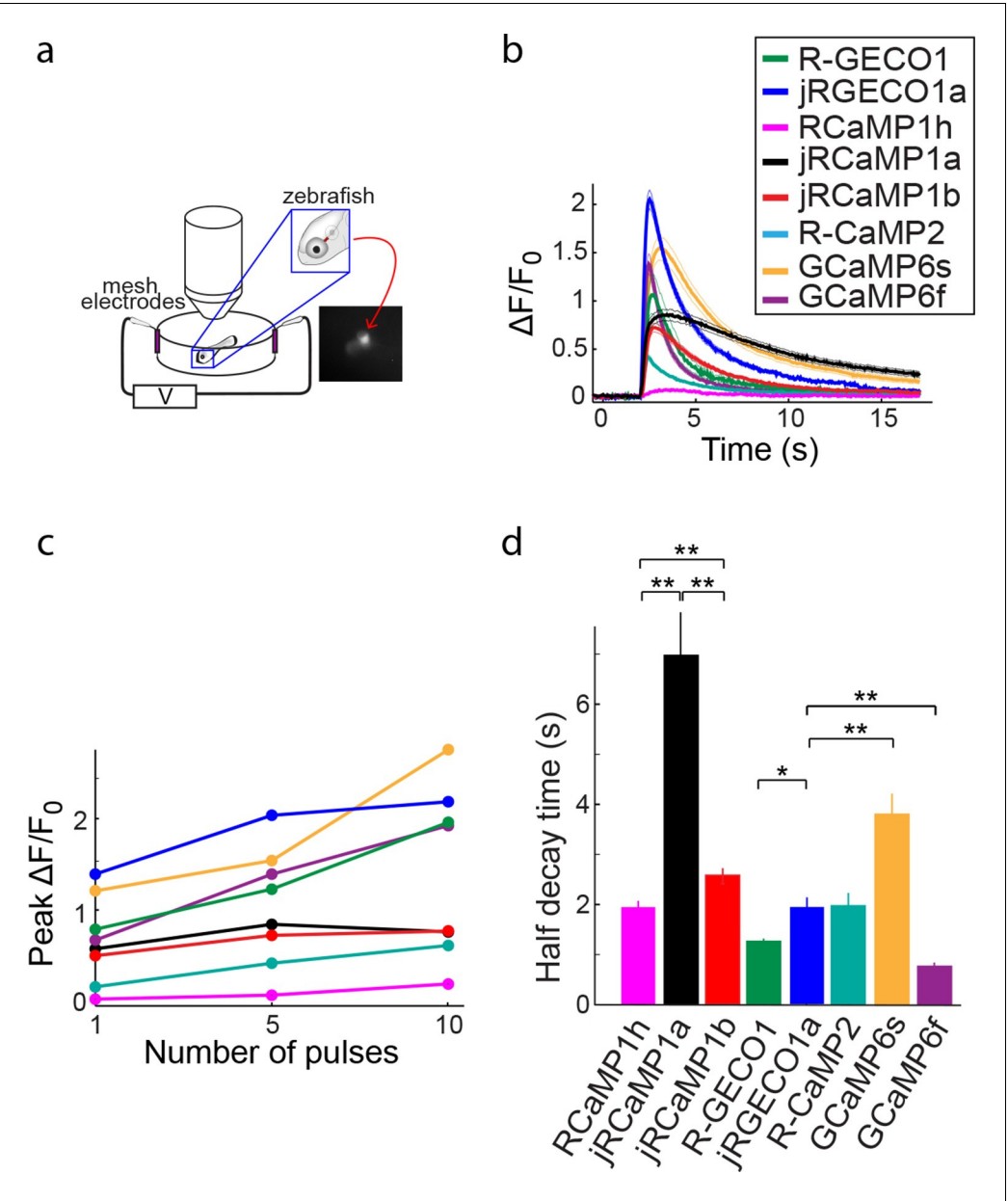

**Figure 8.** Imaging activity in the zebrafish trigeminal neurons with red GECIs. (a) Schematic representation of zebrafish trigeminal neurons assay. Zebrafish larvae (3–4 days post fertilization) were paralyzed, embedded in agarose, and stimulated with electrodes (20 ms pulses; 1, 5, and 10 pulses at 20 Hz; Materials and methods). (b) Response transients to five-pulse stimulus (mean ± s.e.m; n=5 fish for R-GECO1; 7, jRGECO1a; 6, R-CaMP2; 6, RCaMP1h; 7, jRCaMP1a; 6, jRCaMP1b; 6, GCaMP6s; 6, GCaMP6f). Response amplitudes were 7- and 8- fold higher for jRCaMP1a and jRCaMP1b than RCaMP1h, respectively, and 4-fold higher for jRGECO1a than R-CaMP2 (Wilcoxon rank-sum test, p<0.01). (c) Averaged peak ΔF/F₀ (same fish as in b) in response to one, five, and ten pulses stimuli. (d) Half decay time for different red and green GECIs (10 pulses stimulus, mean ± s.e.m., same fish as in b; *, p<0.05; **, p<0.01, Wilcoxon rank sum test).

decay. jRGECO1a and jRCaMP1b showed the best performance in tracking the 1 s stimuli (*Figure 9b,c*), while RCaMP1h, jRCaMP1a, and R-GECO1 showed smaller signal modulations; the dim fluorescence of R-CaMP2 made it less useful. AWC neurons, which are tonically active and inhibited by odor, were exposed to a one minute pulse of 92 µM isoamyl alcohol alternating with buffer (*Figure 9d*). jRGECO1a and jRCaMP1b robustly detected the three known components of the odor

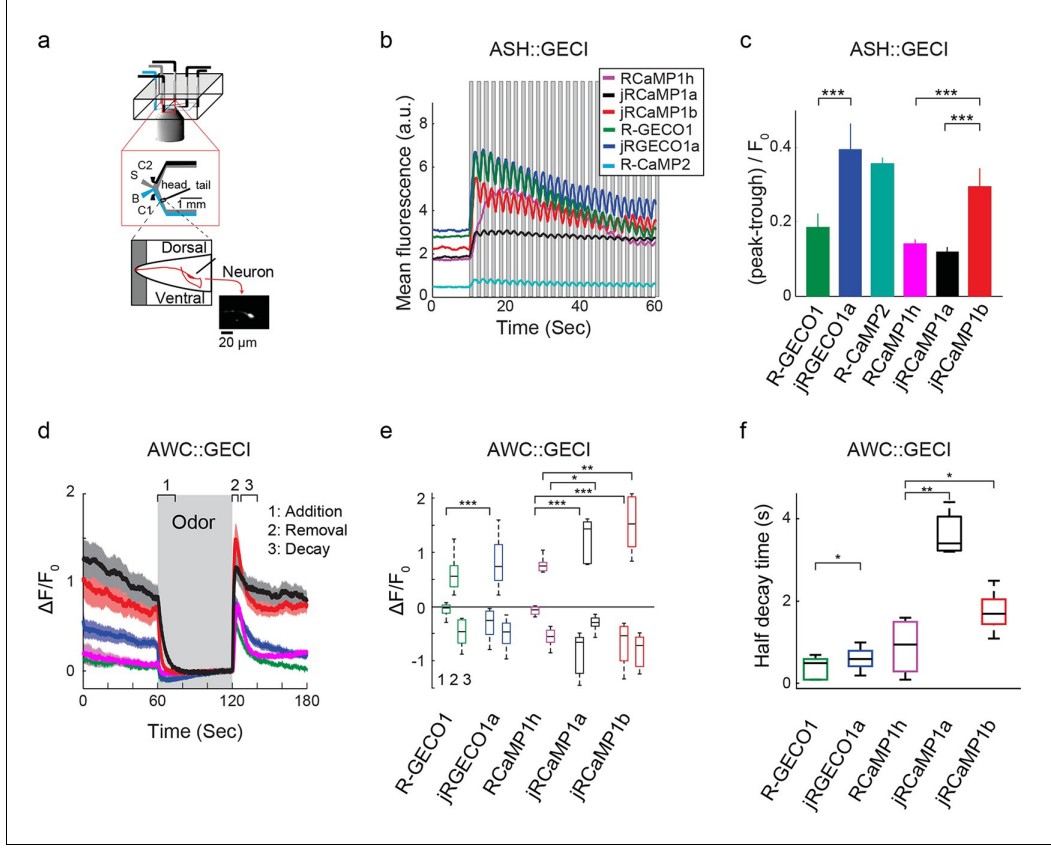

**Figure 9.** Imaging activity in the *C. elegans* ASH and AWC neurons with red GECIs. (a) Schematic representation of the *Caenorhabditis elegans* imaging assay. A paralyzed animal is restrained in a microfluidic device with its nose exposed to a fluid channel. Delivery of stimulus (S) or buffer (B) is controlled indirectly by alternating side streams from control channels C1 and C2. (b) Mean fluorescence transients in ASH neurons in response to 1 s pulses of 1 M hyperosmotic glycerol (mean across worms, n=10 worms for jRGECO1a; 8, jRCaMP1a; 9, jRCaMP1b; 8, R-GECO1; 12, R-CaMP2; 9, RCaMP1h). (c) Quantification of signal modulation across 1 s glycerol pulses (mean peak to trough difference over all 2 s cycles, divided by the signal average fluorescence during the first 10 s in buffer (mean ± s.e.m., same data as in b). Asterisks represent a significant difference (Wilcoxon rank sum test; *, p<0.05; **, p<0.01; ***, p<0.001). (d) Mean fluorescence transients in AWC neurons in response to a one minute exposure to 92 μM isoamyl alcohol (n=19 worms for jRGECO1a; 7, jRCaMP1a; 8, jRCaMP1b; 15, R-GECO1; 6, RcaMP1h). Epochs 1–3 are quantified in **e** and **f**. (e) Comparison of fluorescence changes in AWC during epochs 1 (odor addition), 2 (odor removal) and 3 (return to baseline), same data as in **d**. Each box corresponds to the 25th to 75$^{th}$ percentile of the distribution ($q_1$ and $q_3$ respectively), whisker length is up to the extreme data point or $1.5_l$ ($q_3-q_1$). (f) Comparison of decay rates in AWC upon odor addition (epoch 1), same data as in **d**.

response: odor-induced calcium decreases, a calcium overshoot on odor removal, and a return to baseline (*Figure 9e*). R-GECO1 and RCaMP1h, which have weaker affinity for calcium, did not detect the odor-induced calcium decrease well, whereas jRCaMP1a truncated the overshoot after odor removal (*Figure 9d,e*) and had the slowest kinetics (*Figure 9f*). In summary, jRCaMP1b performed best in detection of graded calcium signals, whereas jRGECO1a had advantages in response speed.

## Discussion

GFP-based GECIs are well-established tools for studying neuronal activity across many model organisms. Current state-of-the-art green GECIs, such as GCaMP6, have sufficient sensitivity to detect single action potentials in diverse cell types and even detect activity of single synapses. They can be targeted to specific cell types and subcellular compartments. Red GECIs can potentially be used in a similar manner with additional advantages. For example, they suffer less from tissue scattering and

absorption. In addition, rhodopsin-based optogenetic tools have substantial absorption in the blue region of the wavelength spectrum, which overlaps with the excitation spectrum of GFP. Red GECIs can be imaged without exciting ChR2. We developed red GECIs that rival best-of-class green GECIs in terms of sensitivity for detecting neural activity, thereby closing a significant performance gap between red and green GECIs.

The new mApple-based jRGECO1a and the mRuby-based jRCaMP1a and jRCaMP1b are all improved several-fold compared to their parent indicators. jRGECO1a exhibits similar performance to the GCaMP6 indicators. jRCaMP1a and jRCaMP1b are less sensitive, but do not show photo-switching after illumination with blue light, making them suitable for experiments that combine calcium imaging and optogenetics. jRCaMP1a and jRCaMP1b are also slightly redder than jRGECO1a. This improved performance will enable a wide range of applications for red GECIs, for example, dual color imaging, where green and red GECIs are used in parallel to study the relation between two components of a neural circuit.

Red GECIs will be useful for all-optical neurophysiology experiments, especially when there is spatial overlap between the stimulated and imaged parts of the neural circuit. The large spectral separation between ChR2 activation spectrum and the red GECI excitation spectrum (*Akerboom et al., 2013*) is preferable in situations when using a red-shifted light sensitive ion channel and a green GECI produces cross-talk (*Rickgauer et al., 2014*; *Packer et al., 2015*). The two-photon absorption spectra of red GECIs peak around 1040 nm, which overlaps with the output of cost-effective and powerful fiber lasers.

Red GECIs still face challenges. First, all tested red GECIs show smaller (3- to 4-fold less) maximal fluorescence changes upon calcium binding than the GCaMP6 indicators. This limits the overall dynamic range of red GECIs. Second, the relatively low absorption cross-section of the red GECIs (*Figure 2—figure supplement 1*) further reduces the achievable signal-to-noise ratio for in vivo imaging. Third, red GECIs show complex intracellular behavior, such as accumulation in lysosomes (jRGECO1a) with long-term expression in the mouse brain. The presence of green fluorescence in all tested GECIs suggests the existence of multiple protein species, some of which do not produce functional signals (*Figure 4—figure supplement 2–3*). These issues seem to be related to the red fluorescent protein component of the sensor (see, for example, *Supplementary file 1* in *Cai et al., 2013*). Therefore, engineering red GECIs to produce less non-productive fluorescence might require replacing or re-engineering the RFP itself (*Moore et al., 2012*), a direction that we plan to pursue in the future.

## Materials and methods

All surgical and experimental procedures were in accordance with protocols approved by the HHMI Janelia Research Campus Institutional Animal Care and Use Committee and Institutional Biosafety Committee.

### Neuronal culture screen

GCaMP indicators are naturally excluded from the nucleus, presumably due to a cryptic nuclear exclusion sequence (NES) motif. This gives rise to the characteristic annular labeling. Both RCaMP and R-GECO are expressed throughout the cell, including the nucleus. GECI responses are larger and faster in the cytoplasm than in the nucleus (unpublished data). We therefore included an NES motif to restrict expression to the cytoplasm. For R-CaMP2, derived from R-CaMP1.07 (*Ohkura et al., 2012*; *Inoue et al., 2015*), it appears that the C-terminal F2A sequence functions as an NES and therefore no additional NES was added.

Mutant red GECIs were made using mismatched oligonucleotides and cloned into an expression vector using gene assembly. The expression vector contained a human synapsin-1 (SYN1) promoter for neuronal-specific expression, a nuclear export signal (N-terminal MLQNELALKLAGLDINKTG, derived from the cAMP-dependent protein kinase inhibitor alpha subunit), a red GECI variant, an internal ribosome entry site, a nuclearly-targeted GFP, and a woodchuck hepatitis virus post-transcriptional regulatory element (WPRE) to augment expression. Amino acid positions are numbered in jRGECO1 and jRCaMP1 variants (*Supplementary files 1–2*) starting from GGSHHHHHHGMASM... (1–14...). NES positions are excluded from the numbering, to correspond with published numbering of the scaffolds.

Red GECI variants were expressed after transfection by electroporation into rat primary hippo-campal neurons (P0) using the Lonza Nucleofector system (P3 Primary Cell 96-well Kit, Lonza, Walkersville, MD). Transfected neurons were plated in glass-bottom 96-well plates (MatTek, Ashland, MA) and cultured as described (*Wardill et al., 2013*).

For stimulation and imaging, synaptic transmission was inhibited with glutamate and GABA receptor antagonists to make calcium changes solely dependent on opening of voltage sensitive cal-cium channels. APs (83 Hz) were evoked by field stimulation as described (*Wardill et al., 2013*), except that stimulation and imaging was in 96-well plates with modified electrodes. Illumination was provided by a warm white LED and TxRed (7mW at the sample plane, excitation: 540–580 nm; dichroic: 585 nm long-pass; emission: 593–668 nm) and GFP (excitation: 450–490 nm; dichroic: 495 nm long-pass; emission: 500–550 nm) filter sets. Imaging conditions and analysis were similar to prior experiments (*Chen et al., 2013b*). For photoswitching measurement in jRGECO1a-expressing neu-rons (*Figure 2—figure supplement 4*), we added blue light stimulation (up to 33 mW) by using a blue LED and replacing the green excitation filter with a 561 nm shortpass filter (Semrock, Rochester, NY).

## Protein expression and purification

Sensors were purified, and calcium titrations were performed as described (*Chen et al., 2013b*). Red fluorescence was measured with excitation at 570 nm (5 nm bandpass) and emission 600 nm (5 nm bandpass). For $k_{off}$ measurements, protein samples in 100 μM calcium were rapidly mixed with a solution of 10 mM EGTA in a stopped-flow device (Applied Photophysics, UK) coupled to a fluorom-eter with 560 nm excitation/580 nm emission filters (20 nm bandpass, Varian). Solutions were buff-ered with 50 mM MOPS pH 7.2, 100 mM KCl at room temperature. Single exponential curves were fitted to the fluorescence decay. pKa was determined as described (*Chen et al., 2013b*). Chromo-phore extinction coefficients ($\varepsilon_{apo}$ and $\varepsilon_{sat}$) were measured as described (*Akerboom et al., 2012*). Values are mean ± standard deviation, except where indicated, for independently purified protein samples.

## Spectroscopy of purified proteins

Purified proteins were characterized in 30 mM MOPS, 100 mM KCl, pH 7.2 (calcium calibration buffer C3008MP, ThermoFisher, Waltham, MA) containing either 10 mM CaEGTA (calcium buffer) or 10 mM EGTA (EGTA buffer). Absorption spectra were taken on a UV/VIS spectrometer (Lambda 35, PerkinElmer, Waltham, MA). Fluorescence emission and excitation spectra were measured on a LS-55 fluorimeter (Perkin-Elmer) with 5 nm slits, and excitation/emission wavelengths of 555 nm/ 600 nm. Absolute quantum yields were measured using a Quantaurus-QY integrating-sphere spec-trometer (model C11374, Hamamatsu, Japan). Dim solutions (*e.g.* jRGECO1a or GCaMP6s in EGTA buffer) were not measurable on the Quantaurus, so relative quantum yields were determined from measured absorption in the UV/VIS spectrometer and integrated fluorescence measured in the fluo-rimeter, in comparison to the reference fluorophores mCherry (measured absolute QY of 0.22) or fluorescein (QY =0.93 in pH 9.5 borate buffer) for either red or green GECIs. Extinction coefficients for RGECO variants were determined using the alkali denaturation method, assuming the denatured extinction coefficient for RGECO variants is the same as denatured mCherry, 37,000 M$^{-1}$cm$^{-1}$ at 455 nm at pH 13 (*Zhao et al., 2011*). For RCaMP variants, the denatured absorption spectra is compli-cated by an absorbance peak at 383 nm that first appears with increasing pH at pH 11.5, and is irre-versibly formed. So for RCaMP variants we used fluorescence correlation spectroscopy (FCS), described below, to quantify concentration by counting fluorophores (*Haustein and Schwille, 2003*), using mCherry as a concentration reference. FCS was performed on 10–100 nM protein solu-tions using 1060 nm excitation at low laser power (1–3 mW) to determine the autocorrelation func-tion G(τ) of the fluorescence signal. The number of fluorophores inside the beam volume is found from 1/G(0). The beam volume was found from identical FCS measurements on the reference protein mCherry at known concentration. For consistency, jRGECO variants were measured using the two methods and results were similar.

## Widefield fluorescence measurement of photoswitching and phoobleaching

Measurements were carried out on aqueous droplets of purified protein isolated in octanol (*Kremers et al., 2009*), and sandwiched between two coverslips. We used an epi-illumination microscope (AxioImager, Zeiss, Germany) with a 20x 0.8 NA air objective (Zeiss). For photoswitching experiments co-linear laser excitation was provided to the microscope at 488 nm (Sapphire 488, Coherent, Santa Clara, CA) and 561 nm (Sapphire 561, Coherent) using a beam-combining dichroic, and computer-controlled shutters determined exposure sequence. Fluorescence collected by the objective passed through dichroic and bandpass filters (Di02-R561 and 625/90, Semrock), and was detected by a fiber-coupled avalanche photodiode (SPCM-AQRH14, Pacer, UK). Laser power was measured at the output of the objective, and laser beam area in the focal plane was imaged and measured using a CCD camera (CoolSnap EZ, Photometrics, Tucson, AZ), in order to determine the laser intensity. Photobleaching experiments were done with the setup above and either 561 nm laser excitation (4.2 W/cm$^2$ at sample) or Hg-lamp excitation (550/25 bandpass, Semrock) yielding 5.6 W/cm$^2$ at the sample.

## Two-photon spectroscopy

Two-photon spectra and FCS were performed as previously described (*Akerboom et al., 2013*). Briefly, protein solutions in coverslip-bottomed dishes were measured on an inverted microscope using a 60x 1.2 NA water-immersion objective (Olympus). Near-infrared laser excitation over the range of 700–1500 nm was provided by either a Ti:sapphire laser (Chameleon Ultra II, Coherent) or an OPO (Chameleon Compact OPO, Coherent). An overlap range 1000–1080 nm was used to connect the fluorescence spectra generated from the two different excitation sources. The Ti:sapphire beam was long-pass filtered (715/LP, Semrock) to remove contaminating visible red spontaneous emission emanating from the Ti:sapphire laser cavity. A single dichroic mirror (675DCSPXR, Omega, Brattleboro, VT) was used to bring the Ti:Sapphire or OPO beams to the objective. Fluorescence collected from protein samples, filtered by the dichroic mirror and a 720/SP short-pass and a 625/90 bandpass filter (Semrock), was detected by an avalanche photodiode detector (SPCM_AQRH-14, Excelitas, Waltham MA). Output pulses from the detector were fed to an autocorrelator (Flex03LQ, Correlator.com, Bridgewater, NJ), and the system operated under computer control with automated data acquisition.

Two-photon action spectra were determined from excitation spectra of 1 micromolar protein solutions obtained at low laser power (0.5 mW), and comparing these to spectra measured under identical conditions for the reference dyes fluorescein and rhodamine B, for which we used published 2-photon cross section data (*Xu and Webb, 1996*; *Makarov et al., 2008*) to determine action spectra of the red GECIs. This setup was also used to perform FCS in order to measure chromophore concentration at low laser power, and two-photon peak brightness (*Mütze et al., 2012*) at high laser power. Peak brightness is the experimentally determined maximum rate of fluorescence per molecule for a given focusing geometry and is a measure of the brightness and photostability of a fluorophore under 2-photon excitation. To determine peak brightness, 20–100 nM solutions of protein in calcium buffer were illuminated at either 1070 nm or 1140 nm at increasing laser power, where at each power setting the mean fluorescence rate F and the mean number of fluorophores N in the laser beam volume were determined by FCS (*Mütze et al., 2012*). The 2-photon peak brightness was then found from the maximum value of the quantity F/N as the laser power was increased.

## Two-photon bleaching measurements

Two-photon bleaching measurements of green and red GECIs were acquired from isolated aqueous droplets of protein using a resonant-galvo scanning 2-photon microscope (MPM-200, Thorlabs, Newton, NJ). The microscope was equipped with a 40x 1.15 NA water immersion objective (Nikon), a primary dichroic (680–1600 nm longpass filter, Thorlabs), and a secondary dichroic (FF562-Di03, Semrock) with green (530/43, Semrock) and red (605/70, Chroma, Bellows Falls, VT) filters each followed by GaAsP PMTs (H7422PA-40, Hamamatsu). Laser excitation was at 1000 nm or 1070 nm for red GECIs, and 940nm for GCaMP6s. To completely bleach the protein droplets (thickness, 5 μm) repetitive z-stacks were taken. Beam scan area was 160 μm x 160 μm, and the scan rate was 8 frames/sec. Light intensity was kept constant across measurements (*Figure 2—figure supplement*

2). However, due to the use of different wavelengths, the laser spot size, and therefore the intensity profile, was slightly different across these measurements.

## AAV-mediated GECI expression in mouse V1 neurons

Constructs used to produce AAV included pGP-AAV-SYN1-red GECI-WPRE and the cre-recombinase-activated construct pGP-AAV-FLEX-SYN1-red GECI-WPRE. Virus (titer: ~0.7-3x10$^{13}$ genomes/ml, R-CaMP2 promoter was CaMKII) was slowly injected (25–30 nl over 5 min) through a thinned skull into the primary visual cortex (1–2 injection sites at variable depths; 2.7 mm lateral and 0.2 mm anterior to lambda suture; 250 µm deep for L2/3 imaging, 250 µm and 450 µm for L4 imaging, 500 µm for L5 imaging, and 650 µm and 900 µm for L6 imaging).

## Cranial window implantation

Sixteen to 180 days after virus injection, mice were anesthetized using isoflurane (3% for induction, 1.5–2% during surgery) and a 2–3 mm circular craniotomy was made above V1 (centered around the injection site). The craniotomy was covered by 1% agarose (UltraPure Agarose, ThermoFisher). A round 3 mm glass coverslip (#1 thickness, Warner Instruments, Hamden, CT ) was cemented to the skull to reduce motion of the exposed brain (*Huber et al., 2012*). A custom titanium head post was fixed to the skull using black dental cement (Contemporary Ortho-Jet, Land Dental, Wheeling, IL). For simultaneous imaging and cell-attached recording, the exposed brain was covered with a thick 1% agarose layer, and partially covered with a D-shape coverslip. The animal was then placed under a microscope on a warm blanket (37°C) and kept anesthetized using 0.5% isoflurane and sedated with chlorprothixene (20–40 µl at 0.33 mg/ml, i.m.) (*Niell and Stryker, 2008*).

## V1 imaging

Imaging was performed with a custom-built 2-photon microscope equipped with a resonant scanner. The light source was an Insight DS Dual 120 femtosecond-pulse laser (Spectra-Physics, Santa Clara, CA) running at 1040 nm and 1100 nm for jRGECO1a and jRCaMP1 indicators, respectively. The objective was a 16x 0.8 NA water immersion lens (Nikon, Japan). Images were acquired using Scan-Image 5 (vidriotechnologies.com) (*Pologruto et al., 2003*). For L2/3, L4, and L5 imaging, functional images (512x512 pixels, 250x250 µm$^2$) were collected at 15 Hz. Typical laser powers were 15–60, 30–75, and 50–100 mW at the front aperture of the objective for L2/3, L4, and L5 imaging, respectively. For L6 imaging, functional images (512x512 pixels, 165x165 µm$^2$ or 125x125 µm$^2$) were collected at 5 Hz. Laser power was 100–150 mW at the front aperture of the objective lens.

For dual-color imaging we used green and red detection channels (525/50 nm and 600/60 nm filters respectively, separated by a 565 nm LP dichroic mirror, Chroma). Bleedthrough of the red signal into the green channel was negligible, and penetration of green signal into the red channel was corrected by linear unmixing.

## Visual stimuli

Moving grating stimuli were generated using the Psychophysics Toolbox (*Brainard, 1997*; *Pelli, 1997*) in MATLAB (Mathworks, Natick, MA). Each stimulus trial consisted of a 4 s blank period (uniform gray at mean luminance) followed by a 4 s drifting sinusoidal grating (0.05 cycles per degree, 1 Hz temporal frequency). Eight drifting directions were used, separated by 45°, and 5 trials were recorded for each direction, giving a total of 40 stimulus trials per recording session (320 s recording time). The gratings were presented with an LCD monitor (30 x 40 cm), placed 25 cm in front of the center of the right eye of the mouse. The monitor subtended an angle of ± 38° horizontally and ± 31° vertically around the eye of the mouse. For experiments with cell-attached recording (*Figure 4*), pipette access required the use of a smaller LCD monitor (12 x 16 cm) placed 10 cm in front of the right eye. During simultaneous imaging and electrophysiology, the optimal grating stimulus was repeatedly played, but the contrast of the stimulus grating was adjusted online to record variable spike rates. To minimize stimulus contamination in the red detection channel, a green colored plexiglass filter (500–555 nm FWHM) was placed in front of the mouse eye.

## Analysis of V1 functional imaging

Data from mice expressing GECIs for 16 to 64 days were used for the following analysis (*Figure 3c–f*). Mechanical drift in the imaging plane was corrected using the TurboReg plug-in in ImageJ (*Thévenaz et al., 1998*). All remaining analyses were performed in MATLAB. Regions of interest (ROIs) corresponding to identifiable cell bodies were selected using a semi-automated algorithm (*Akerboom et al., 2012*; *Chen et al., 2013b*). Depending on a neuron's appearance, annular or circular ROIs were placed over the cytosolic region of each cell. The fluorescence time course was measured by averaging all pixels within the ROI, after correction for neuropil contamination (*Kerlin et al., 2010*). The neuropil signal $F_{neuropil}(t)$ surrounding each cell was measured by averaging the signal of all pixels within a 20 μm circular region from the cell center (excluding all somata). The fluorescence signal of a cell body was estimated as

$$F_{cell\_true}(t) = F_{cell\_measured}(t) - r.F_{neuropil}(t),$$

with r = 0.7. Neuropil correction was applied only to cells with baseline fluorescence ($F_0$) signal stronger than the surrounding neuropil signal by more than 3%; other cells (approximately 15–20%) were excluded from the analysis because $F_0$ could not be reliably estimated. After neuropil correction, the $\Delta F/F_0$ of each trial was calculated as $(F-F_0)/F_0$, where $F_0$ was averaged over a 1 s period immediately before the start of grating stimulation. Visually responsive neurons were defined as cells with $\Delta F/F_0$ >0.05 during at least one stimulus period, and using ANOVA across blank and eight direction periods (p<0.01).

We calculated the decay time of fluorescence after the end of the preferred stimulus. For each cell we averaged responses from five trials; baseline fluorescence and standard deviation were calculated from 1 s before the start of the stimulus. Only cells with fluorescence response peaks with >4 times the standard deviation of the baseline during the last 1 s of the stimulus were analyzed. The time required for each trace to reach half of its peak value (baseline fluorescence subtracted) was calculated by linear interpolation. The same cells were used for plotting the average response showed in *Figure 3c*. Because each cell responded at slightly different times, depending on its receptive field structure, each trace was shifted to align maxima. For display purposes, traces were smoothed with a 3 sample moving average kernel.

The orientation selectivity index (OSI, *Figure 3—figure supplement 2*) was calculated for visually responsive cells (*Niell and Stryker, 2008*). First, the preferred orientation ($\theta_{pref}$) of the cell was determined as the stimulus evoking the greatest response. The orientation tuning curve was constructed by measuring the mean $\Delta F/F_0$, averaged over the stimulus period, for each orientation. We then fitted the tuning curve with the sum of two Gaussians centered on $\theta_{pref}$ and $\theta_{pref}+\pi/2$, both with width σ, amplitudes A1 and A2, and a constant baseline B. The OSI was defined as OSI $=(R_{pref} - R_{ortho})/(R_{pref}+R_{ortho})$, where $R_{pref}$ and $R_{ortho}$ are the response amplitudes at the preferred ($\theta pref$) and the orthogonal orientation ($\theta_{pref}+\pi/2$) respectively.

## Simultaneous electrophysiology and functional imaging in V1

In vivo cell-attached recordings were performed using glass pipettes (~7–12 MΩ) filled with (in mM): 125 NaCl, 5 KCl, 10 glucose, 10 HEPES, 2 CaCl2, 2 MgSO4, and 0.1 Alexa Fluor 488; pH 7.4). Signals were amplified using an AxoPatch 200B amplifier (Molecular Devices, Sunnyvale, CA), filtered at 5 kHz, and digitized at 10 kHz. Spikes were recorded using current clamp mode. The frame trigger pulses of ScanImage 5 were also recorded and used offline to synchronize individual frames to electrophysiological recordings. After establishment of a low-resistance seal (15–50 MΩ), the stimulus orientation was quickly optimized for individual neurons using recorded spikes. The optimal grating stimulus was repeated with variable contrast to acquire a range of spiking rates.

Images (512x512 pixels, typically 40x40 μm$^2$) were acquired at 30 Hz and 15 Hz for jRGECO1a and jRCaMP1a, respectively. Ring-shaped ROIs were placed over the cytosolic regions of the cells, and neuropil contamination was subtracted. To quantify the fidelity of single AP detection (*Figure 4e*), we identified single AP events isolated from other APs by >1 s, and 2 APs within a time window of 100 and 150 ms for jRGECO1a and jRCaMP1a respectively, isolated from other APs by >1 s. Noise traces were taken from periods without APs. $F_0$ values were calculated as the average of 10 frames before the first AP, and peak $\Delta F/F_0$ was calculated as the maximum of the $\Delta F/F_0$ trace after the first spike firing, or the maximal $\Delta F/F_0$ noise value for the noise traces. For d-prime

calculation and receiver operating characteristic (ROC) curves, response templates for various APs were calculated by averaging response traces. Unit vectors were calculated from the templates by subtracting means and normalizing the vectors. The measured n-AP responses were projected on the unit vectors, where n indicates the number of APs in a time bin, as well as noise traces (*Chen et al., 2013b*). The scalar results of projecting the response and noise traces on the unitary response templates were used to calculate d-prime and ROC curves.

Time windows in which the APs were fired varied with the number of spikes and the GECI (jRGECO1a: 100 ms for 2APs and increasing by 25 ms for each additional AP, up to 250 ms for 8APs; jRCaMP1a: 150 ms for 2APs and increasing 50 ms for each additional AP, up to 350 ms for 6–8 APs). Time bins were chosen to reflect the accumulation of calcium signal for large number of APs (longer for the slower jRCaMP1a/b, shorter for the faster jRGECO1a). Half rise and decay times were calculated from 2 AP responses. Only isolated responses (without APs > 2 s before the recorded spikes) were used. An exponential decay curve was fitted to each trace. Only traces with small fitting error were chosen (16 traces for both indicators).

## Fixed tissue analysis and immunostaining

Mice were deeply anesthetized with isoflurane and transcardially perfused with 10 ml 1X Dulbecco's phosphate-buffered saline (DPBS, ThermoFisher), followed by 50 ml 4% paraformaldehyde in 0.1 M phosphate buffer. After perfusion, the brains were removed and post-fixed overnight at 4°C. The brains were embedded in 5% agarose in DPBS, and cut into 50 µm thick coronal sections with a vibratome (Leica VT 1200S, Germany). Because DPBS contains saturating calcium (0.9 mM) red GECI brightness is maximal. Sections were coverslipped with Vectashield mounting medium (H-1400, Vector laboratories). Confocal images (LSM 710, Zeiss) were collected using a 20x 0.8 NA air objective. Emission spectra were characterized using a lambda scanning mode of the confocal microscope, and were corrected for background emission.

For LAMP-1 immunostaining, brains were perfused, fixed (4 hr post-fixed, room temperature), and sectioned as described above. Sections were soaked and washed twice with DPBS, then blocked in blocking buffer (2% BSA, 0.4% Triton X-100 in DPBS) for 1 hr at room temperature. Sections were then incubated in anti-LAMP-1 antibody conjugated to Alexa Fluor 488 (clone 1D4B, diluted 1:200 in blocking buffer, Santa Cruz Biotechnology, Dallas, TX) overnight at 4°C, and then washed 3 times with 0.1% Triton X-100 in DPBS, and washed again with DPBS. Finally, sections were mounted and coverslipped with Vectashield anti-fade mounting medium (H-1500, Vector laboratories, Burlingame, CA).

## *Drosophila* NMJ imaging

We made $w^{1118}$;; $PBac\{20XUAS\text{-}IVS\text{-}GECI\text{-}p10\}VK00005$ transgenic flies and crossed them with $w^{1118}$;; $R57C10\text{-}Gal4$ in VK00020, $R57C10\text{-}Gal4$ in VK00040 pan-neuronal driver line. Improved expression allows lower excitation light dosage and helps to prevent photobleaching. The NMJ assay is similar to the one used previously (*Chen et al., 2013b*). Briefly, actively crawling female 3rd instar larvae were dissected under minimum illumination intensity. Type Ib boutons on muscle 13 from segment A3-A5 were wide-field imaged in HL-6 saline while corresponding axons were electrically stimulated with a suction electrode driven by a customized stimulator. Temperature and pH were monitored during imaging. A mercury lamp (X-CITE exacte, Excelitas) light source was used for excitation and power of less than 5 mW at the objective front aperture was used. The light intensity was calibrated so that no significant photobleaching was detected during each trial. The filters for jRGECO1 and jRCaMP1 imaging were: excitation: 543/22 nm; dichroic: 562 nm; emission: 593/40 nm. Emitted light was collected on an EMCCD cooled to -70°C at 30 fps. Data were analyzed in MATLAB (MathWorks).

## Zebrafish trigeminal neurons imaging

Mitfa$^{-/-}$ (nacre) zebrafish were maintained under standard conditions at 28°C and a 14:10 hr light: dark cycle. Embryos (1–2 cell stage) were injected with 20 ng/µl DNA plasmids encoding the red GECI variants under the control of the (near) pan-neuronal elavl3/HuC promoter (elavl3:GECI), and 40 ng/µl Tol2 transposase mRNA diluted in E3 medium with 0.025% Phenol Red. Three and four day post-fertilization embryos showing expression in isolated trigeminal neurons were treated with 1

mg/ml bath-applied α-bungarotoxin for 30 min to block tail movements. Larvae were mounted sideways in 1.5% low melting temperature agarose, and imaged using a microscope with a 20×, NA=1 objective lens (Olympus, Japan) and an sCMOS camera (Hamamatsu Orca Flash 4). Trains of 1, 5 and 10 field stimuli (20 ms each; 20 Hz) were applied using two mesh electrodes located inside the bath and a stimulator (Grass SD9, Warwick, RI). Stimulation voltage was calibrated to elicit an identifiable response to a single pulse in GCaMP6f expressing neurons. One spike in a trigeminal neuron has previously been shown to be sufficient to elicit tail movement (*Douglass et al., 2008*). Image acquisition and stimulus control were handled using Hamamatsu HCImage (4.2.5.0) software. ROIs were selected manually, and data was analyzed using MATLAB (MathWorks).

### *Caenorhabditis elegans* ASH and AWC neurons imaging

Red GECIs were expressed in the *C. elegans* ASH and AWC sensory neurons under the *sra-6* and *str-2* promoters, respectively. Animals were restrained in custom-built microfluidic chambers (*Chronis et al., 2007*) in S Basal buffer (*Brenner, 1974*), and paralyzed with 1 mM tetramisole hydrochloride (Sigma-Aldrich, St. Louis, MO) during data acquisition to reduce movement. For ASH imaging, 1 M glycerol was delivered to the nose of the animal in alternating one second intervals for 4 min as in (*Kato et al., 2014*). For AWC imaging, 92 µM isoamyl alcohol was delivered for one minute after one minute of exposure to buffer. Worms were illuminated with a solid-state lamp (Lumencor SOLA-LE, Beaverton, OR, 560/40 nm excitation filter), and fluorescence images (630/75 nm emission filter) were collected at 10 fps using a 40x, NA=1.4 objective (Zeiss) and an EMCCD camera (Andor iXon3, Metamorph Software, 250 x 140 pixels, covering 104 µm x 58 µm).

### Reagent distribution

DNA constructs, AAV particles and *Drosophila* with red GECIs variants were deposited for distribution at Addgene (http://www.addgene.org), the University of Pennsylvania Vector Core (http://www.med.upenn.edu/gtp/vectorcore) and the Bloomington *Drosophila* Stock Center (http://flystocks.bio.indiana.edu), respectively.

Cell-attached recording and functional imaging data were deposited at CRCNS.org (doi: 10.6080/K0W37T87).

## Acknowledgements

We thank R Campbell for an advice on red GECI lysosome accumulation, J Akerboom for reagents, K Ritola, J A Rouchard, S Coffman, B Sharp, and K Hibbard, for technical assistance.

## Additional information

#### Competing interests
DSK, ERS, LLL and KS have applied for a patent on materials and methods related to the red GECI variants (application number US 14/974,483). The other authors declare that no competing interests exist.

## Funding

| Funder | Author |
|--------|--------|
| Howard Hughes Medical Institute | Hod Dana |
| | Boaz Mohar |
| | Yi Sun |
| | Sujatha Narayan |
| | Andrew Gordus |
| | Jeremy P Hasseman |
| | Getahun Tsegaye |
| | Graham T Holt |
| | Amy Hu |
| | Deepika Walpita |
| | Ronak Patel |
| | John J Macklin |
| | Cornelia I Bargmann |
| | Misha B Ahrens |
| | Eric R Schreiter |
| | Vivek Jayaraman |
| | Loren L Looger |
| | Karel Svoboda |
| | Douglas S Kim |

The funders had no role in study design, data collection and interpretation, or the decision to submit the work for publication.

### Author contributions

HD, YS, AG, JJM, DSK, Conception and design, Acquisition of data, Analysis and interpretation of data, Drafting or revising the article; BM, AH, DW, Acquisition of data; SN, ERS, Acquisition of data, Analysis and interpretation of data, Drafting or revising the article; JPH, GT, GTH, Performed protein assays; RP, Acquisition of data, Analysis and interpretation of data; CIB, MBA, VJ, LLL, KS, Conception and design, Analysis and interpretation of data, Drafting or revising the article

### Author ORCIDs

Cornelia I Bargmann, http://orcid.org/0000-0002-8484-0618
Misha B Ahrens, http://orcid.org/0000-0002-3457-4462
Karel Svoboda, http://orcid.org/0000-0002-6670-7362
Douglas S Kim, http://orcid.org/0000-0001-9029-314X

### Ethics

Animal experimentation: All experimental protocols were conducted according to National Institutes of Health guidelines for animal research and were approved by the Institutional Animal Care and Use Committee at Janelia Research Campus (protocol 13-95).

## Additional files

### Supplementary files

• Supplementary file 1. Comprehensive neuronal culture screening results for RCaMP1h variants.

• Supplementary file 2. Comprehensive neuronal culture screening results for R-GECO1 variants.

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
