## [Decision Letter]

Thank you for submitting your work entitled "Sensitive red protein calcium indicators for imaging neural activity" for consideration by *eLife*. Your article has been reviewed by three peer reviewers, and the evaluation has been overseen by a Reviewing Editor and Gary Westbrook as the Senior Editor.

The reviewers have discussed the reviews with one another and the Reviewing Editor has drafted this decision to help you prepare a revised submission.

Summary:

This manuscript describes the development and testing of improved red-shifted genetically encoded calcium indicators (GECIs). For optimization of the parental R-CaMP and R-GECO the team at Janelia Farm employed the same strategy that previously had been successfully used for a new generation of G-CaMPs, which are now widely used. It consists of structure-guided mutagenesis of the indicator proteins, followed by a neuronal screening assay using field stimulation to elicit action potential firing. Screening was followed by a careful and detailed characterization of the new probes in several model organisms in vivo.

The authors convincingly demonstrate that especially the variant jRGECO1a shows greatly improved performance compared to previous red GECIs in many preparations, rivaling state-of-the-art green GECIs. The mRuby-based jRCaMP1a, which could be used for combined optogenetic photostimulation and imaging experiments, shows promise, but still has very slow decay kinetics and a limited dynamic range. It should be noted that some of the bugs of the parental probes that make their use difficult at times have not been fixed: jRGECOs still show photoswitching upon blue light illumination, and all sensors still partition into green and red species, with the green species not being calcium sensitive, buffering calcium without contributing signal (complicating co-use with GCaMPs).

Overall this is a very thorough manuscript, providing extensive data about the new red GECI variants in several preparations. These new calcium indicators and the information provided about them will be of great value and interest to the field. There are some issues which still need to be addressed.

Essential revisions:

1) The green GCaMP6 variants can begin to show adverse effects on neuronal health after long-term expression of several weeks. It is therefore important to assess the degree to which this is a problem for the red GECIs, with and without NES (nuclear export sequences). For example, as in supplementary Figure 7 from Chen et al. 2013 (same group): Do the signals change? Do cell nuclei tend to fill after a long time? Please provide a comparison of biophysical properties and in vivo response properties such as spontaneous activity levels, response reliability and calcium transient shape (orientation selectivity is not a good indicator for the health of a neuron!) after short- and longer-term expression of the red GECIs (e.g. after 3 and 10 weeks).

2) The authors claim that these new red GECIs have "sensitivity comparable to GCaMP6". They also mention in what ways they are different. It would be nice if some of these key comparisons were included in main figures. (a) It would be wise to move Figure 2—figure supplement 1, panel c to a main figure. (b) It would also be helpful to show how the brightness and bleaching compare. To address that: Figure 2—figure supplement 2 – please plot similar measurements from GCaMP6 variants (at least 6s… 6f would be nice to have as well). Finally, (c) could the authors please add statistical information to statements describing increased/decreased performance of specific GECIs in comparison to others, for instance in the last few sentences of the second paragraph of subsection “Red protein calcium indicators in mouse V1”, Figure 3; Figure 7; Figure 8.

3) The authors provide data on photobleaching of the purified proteins in aqueous droplets. It is not clear how this translates to the protein expressed in live tissue. Could the authors please provide information on the extent of photobleaching under realistic in vivo imaging conditions?

4) When estimating spike detection from calcium signals in simultaneous imaging and cell-attached recordings, the authors use a small field-of-view of 40 μm x 40 μm. This is a much smaller field-of-view than those normally used in imaging experiments (250 μm x 250 μm in this study). At the larger zoom the imaged cells will contain many more pixels over which the response is averaged, signal-to-noise will therefore be larger and spike-related transients will be easier to detect. It would be much more useful for the imaging community to perform such "ground-truth" experiments with the experimental settings generally used in the actual imaging experiments, otherwise information such as detection probability of single spikes has only limited applicability. Ideally the authors should provide new electrophysiological ground-truth calibrations to address this issue; but in light of the fact that such experiments are difficult, the authors should at the very least acknowledge the limitations of the way they have performed the calibration. If it is not possible to provide new data, we also suggest some new analysis to address this issue: Figure 5 suggests that peak dF/F is a linear function of the number of spikes, but it actually depends on frequency as well. The authors currently obscure this by using time bins (which aren't even held constant across the two reporters). We suggest doing this analysis and adding it to the report; i.e. simply show that the max dF/F varies significantly with the frequency of the spike train (this is true for GCaMP6 at least). Moreover, Figure 5 should have the individual data points indicated as well. This is a case in which the SD really matters more than the SEM, and thus seeing the spread of the actual data can be informative.

5) Appearance of the green species: Is it present right after onset of expression or a consequence of long term expression? If so, it would be useful to see a time course of green species expression profiles in vivo after viral gene delivery. This could help users to find a suitable time window that offers best conditions for their experiments.

6) How do the authors explain the differing performance of the sensors in the various model organisms? G-CaMP6s appears to clearly outperform jRGECO1 in layer 2/3 neurons of mouse visual cortex, whereas at the *Drosophila* NMJ preparation, jRGECO1 overall appears to do better than GCaMP6s. Given that the NMJ prep has essentially no tissue covering the boutons the brighter green GCaMP6s should have an easy game while in layer 2/3 mouse cortex the higher penetration of red light might favor jRGECO1. Why is this not the case?

7) The authors write, "GECI responses are larger and faster in the cytoplasm than in the nucleus (unpublished data)." Although the "faster" part is consistent with the findings in other labs, it has been observed by others that the signal can sometimes actually be larger in the nucleus than in the cytoplasm. Thus, it would be helpful if the authors could share some of this unpublished data in the supplement.

---

## [Author Response]

Essential revisions:

*1) The green GCaMP6 variants can begin to show adverse effects on neuronal health after long-term expression of several weeks. It is therefore important to assess the degree to which this is a problem for the red GECIs, with and without NES (nuclear export sequences). For example, as in supplementary Figure 7 from Chen et al. 2013 (same group): Do the signals change? Do cell nuclei tend to fill after a long time? Please provide a comparison of biophysical properties and in vivo response properties such as spontaneous activity levels, response reliability and calcium transient shape (orientation selectivity is not a good indicator for the health of a neuron!) after short- and longer-term expression of the red GECIs (e.g. after 3 and 10 weeks).*We studied the performance of red GECIs as a function of time of expression (up to 180 days after injection) (new Figure 3—figure supplement 2). We compared four different parameters: proportion of stimulus responsive cells, proportion of orientation-tuned cells, peak response amplitudes, and response decay times. Previous studies (Tian et al., Nature Methods, 2009, and Chen et al., Nature, 2013) showed that GCaMP accumulation in the nucleus is correlated with changes in these parameters. For the red GECIs there were no nuclear-filled cells under any conditions due to the effects of NES in the red GECI sequences. All parameters were stable as a function of time (with the possible exception of jRCaMP1b).

*2) The authors claim that these new red GECIs have "sensitivity comparable to GCaMP6". They also mention in what ways they are different. It would be nice if some of these key comparisons were included in main figures. (a) It would be wise to move Figure 2—figure supplement 1, panel c to a main figure. (b) It would also be helpful to show how the brightness and bleaching compare. To address that: Figure 2—figure supplement 2 – please plot similar measurements from GCaMP6 variants (at least 6s… 6f would be nice to have as well). Finally, (c) could the authors please add statistical information to statements describing increased/decreased performance of specific GECIs in comparison to others, for instance in the last few sentences of the second paragraph of subsection “Red protein calcium indicators in mouse V1”, Figure 3; Figure 7; Figure 8.*A) We added a new figure (Figure 5) on dual-color imaging using jRGECO1a and GCaMP6s, which includes the former Figure 2—figure supplement 1 panel.

B) We added information on GCaMP6s biophysical properties to Figure 2—figure supplement 2.

C) Statistical information was added to the text and Figure 3, Figure 8, and Figure 9.

*3) The authors provide data on photobleaching of the purified proteins in aqueous droplets. It is not clear how this translates to the protein expressed in live tissue. Could the authors please provide information on the extent of photobleaching under realistic in vivo imaging conditions?*The characterization of photophysical GECI properties was done in a purified protein assay. This provides precise control over the GECI concentration, droplet dimension, and laser power delivered to the protein. Comparable control is difficult to achieve in vivo.

We added qualitative observations on bleaching to the text. We have also added two movies showing fluorescence signal from cells and dendrites over several minutes of imaging.

4) When estimating spike detection from calcium signals in simultaneous imaging and cell-attached recordings, the authors use a small field-of-view of 40 μm x 40 μm. This is a much smaller field-of-view than those normally used in imaging experiments (250 μ

*m x 250 μm in this study). At the larger zoom the imaged cells will contain many more pixels over which the response is averaged, signal-to-noise will therefore be larger and spike-related transients will be easier to detect. It would be much more useful for the imaging community to perform such "ground-truth" experiments with the experimental settings generally used in the actual imaging experiments, otherwise information such as detection probability of single spikes has only limited applicability. Ideally the authors should provide new electrophysiological ground-truth calibrations to address this issue; but in light of the fact that such experiments are difficult, the authors should at the very least acknowledge the limitations of the way they have performed the calibration. If it is not possible to provide new data, we also suggest some new analysis to address this issue: Figure 5 suggests that peak dF/F is a linear function of the number of spikes, but it actually depends on frequency as well. The authors currently obscure this by using time bins (which aren't even held constant across the two reporters). We suggest doing this analysis and adding it to the report; i.e. simply show that the max dF/F varies significantly with the frequency of the spike train (this is true for GCaMP6 at least). Moreover, Figure 5 should have the individual data points indicated as well. This is a case in which the SD really matters more than the SEM, and thus seeing the spread of the actual data can be informative.*We performed the cell-attached recording and fluorescence imaging under favorable conditions in order to get the optimal detection sensitivity for the GECIs (similar to Chen et al., 2013). By subsampling the number of pixels per cell, more realistic experimental conditions can be simulated. The data we acquired will be deposited for free use by other laboratories to achieve that goal (at crcns.org, similar to our GCaMP5 and GCaMP6 data).

A) We added a note to the text that these experiments were done under favorable conditions.

B) The frequency analysis suggested by the reviewer requires a larger database. Data would have to be divided into groups according to the number of spikes in a time bin *and* the burst frequency. We and others in this field have grouped the responses according to the number of spikes in a time bin. Results of the suggested analysis are shown in Figure 10.

C) The time bins we used reflect the different GECI kinetics (longer for slower GECIs). Performing the same analysis with a constant time bin (i.e. 250ms like in Chen et al., 2013) does not change the mean response much, but increases the variability, and reduces the detection sensitivity. We added a clarification to the text.

D) We modified Figure 5 (Figure 6 in the revised manuscript) to show the data distribution.

Author response image 1.**DOI:**
http://dx.doi.org/10.7554/eLife.12727.032

*5) Appearance of the green species: Is it present right after onset of expression or a consequence of long term expression? If so, it would be useful to see a time course of green species expression profiles in vivo after viral gene delivery. This could help users to find a suitable time window that offers best conditions for their experiments.*We now show that jRGECO1a (but not jRCaMP1a/b) accumulates in lysosomes and that this phenomenon is more pronounced during long-term expression in vivo compared to cultured neurons. We also show that this accumulation contributes to lower response amplitudes (Figure 6—figure supplement 1). In addition we added a more detailed characterization of the greenish emission and the change of the jRGECO1a greenish component with time, as the reviewers suggested (Figure 6—figure supplement 3).

*6) How do the authors explain the differing performance of the sensors in the various model organisms? G-CaMP6s appears to clearly outperform jRGECO1 in layer 2/3 neurons of mouse visual cortex, whereas at the Drosophila NMJ preparation, jRGECO1 overall appears to do better than GCaMP6s. Given that the NMJ prep has essentially no tissue covering the boutons the brighter green GCaMP6s should have an easy game while in layer 2/3 mouse cortex the higher penetration of red light might favor jRGECO1. Why is this not the case?* We agree that the red sensors responses in the mouse cortex are lower than what is expected based on the cultured cells assay results. The *Drosophila* NMJ data is similar to the cultured neuron data, and the mouse V1 data appears to be the exception here. The sources of the reduced V1 performance were studied (see response above) and are now presented in Figure 6—figure supplements 1-3, and mentioned in the Discussion. Multiple mechanisms might contribute, including non-productive fluorescence in lysosomes and non-productive green species.

7) The authors write, "GECI responses are larger and faster in the cytoplasm than in the nucleus (unpublished data)." Although the "faster" part is consistent with the findings in other labs, it has been observed by others that the signal can sometimes actually be larger in the nucleus than in the cytoplasm. Thus, it would be helpful if the authors could share some of this unpublished data in the supplement.

In comparisons using red GECIs with and without an NES we consistently find smaller responses without an NES (where sensor is found in both the cytoplasm and nucleus). See for example Figure 11. The data show that responses are smaller and slower in the nucleus. We feel that our description in the text is adequate.

Author response image 2.**DOI:**
http://dx.doi.org/10.7554/eLife.12727.033